# Virus and cell specific HMGB1 secretion and subepithelial infiltrate formation in adenovirus keratitis

Amrita Saha[1,2], Mohammad Mirazul Islam[1,3], Rahul Kumar[4], Ashrafali Mohamed Ismail[1], Emanuel Garcia[2], Rama R. Gullapali[4], James Chodosh[1,2,3], Jaya Rajaiya[1,2]*

1 Department of Ophthalmology, Massachusetts Eye and Ear, Harvard Medical School, Boston, Massachusetts, United States of America, 2 Department of Molecular Genetics and Microbiology, University of New Mexico Health Sciences Center, Albuquerque, New Mexico, United States of America, 3 Department of Ophthalmology and Visual Sciences, University of Ophthalmology and Visual Sciences, University of New Mexico Health Sciences Center, Albuquerque, New Mexico, United States of America, 4 Department of Pathology, University of New Mexico Health Sciences Center, Albuquerque, New Mexico, United States of America

* jrajaiya@salud.unm.edu

## Abstract

A highly contagious infection caused by human adenovirus species D (HAdV-D), epidemic keratoconjunctivitis (EKC) results in corneal subepithelial infiltration (SEI) by leukocytes, the hallmark of the infection. To date, the pathogenesis of corneal SEI formation in EKC is unresolved. HMGB1 (high-mobility group box 1 protein) is an alarmin expressed in response to infection and a marker of sepsis. Earlier studies using a different adenovirus species, HAdV-C, showed retention of HMGB1 in the infected cell nucleus by adenovirus protein VII, enabling immune evasion. Here, using HAdV-D we show cell-specific HMGB1 secretion by infected cells, and provide an HAdV-D specific mechanism for SEI formation in EKC. HMGB1 was secreted only upon infection of human corneal epithelial cells, not from other cell types, and only upon infection by HAdV-D types associated with EKC. Acetylated HMGB1 translocation from the nucleus to the cytoplasm, then to the extracellular milieu, was tightly controlled by CRM1 and LAMP1, respectively. Primary stromal cells when stimulated by rHMGB1 expressed proinflammatory chemokines. In a novel 3D culture system in tune with the architecture of the cornea, HMGB1 released by infected corneal epithelial cells induced leukocytic infiltrates either directly and/or indirectly via stimulated stromal cells, which together explains SEI formation in EKC.

## Author summary

Ocular surface infection by human adenovirus species D causes epidemic keratoconjunctivitis (EKC), a highly contagious form of pink eye. The most significant long-term complication of EKC is corneal subepithelial infiltrate (SEI) formation.

---

**Data availability statement:** All data are available without restriction. Minimal data set is now in the S1 Table.

**Funding:** This work was funded by the National Institutes of Health (R01 EY021558 to JR and JC; R01 EY013124 to JR and JC; R00 EY031373 to MMI; and P20GM121176 to the Autophagy, Inflammation and Metabolism Center at the University of New Mexico) The funders had no role in study design, data collection and analysis, decision to publish, or preparation of the manuscript.

**Competing interests:** The authors have declared that no competing interests exist.

SEI appear as round, grayish white opacities in the corneal stroma just beneath the corneal epithelium, and cause discomfort and blurred vision. SEI represent the clinical manifestation of immune cell migration into the subepithelial region in response to adenovirus infection. However, the mechanism of SEI formation after adenovirus infection of the cornea remains unknown. In our study, we found that a secretory protein, high-mobility group box protein 1 (HMGB1), is specifically released by adenovirus infected human corneal epithelial cells. We show that HMGB1 stimulates proinflammatory responses by corneal stromal cells beneath the epithelium. Using a novel 3D cultured "cornea in a test tube", we show that HMGB1 expression by infected corneal epithelial cells induces immune cell migration to the corneal stroma, similar to SEI formation in adenovirus infected human eyes. This explains the means by which adenovirus infection of the corneal surface epithelium induces inflammation in the underlying corneal stroma. HMGB1 is a viable therapeutic target for preventing the vision-threatening complications of EKC.

## Introduction

As with the HIV epidemic, the COVID-19 pandemic with its high mortality rate brought increased attention to research in viral pathogenesis [1], and also to adenoviruses, the latter principle targets for vaccine development during the pandemic [2,3]. Adenoviruses are highly prevalent causes of infection, and although less likely to cause death, are also associated with epidemics with significant associated morbidity, mortality, and societal costs [4]. Different adenoviruses utilize disparate means of viral entry and generate unique immune response signatures that also differ by target cell type and tissue [5]. The conjunctiva is the mucous membrane lining the entire ocular surface except for the contiguous epithelial cell layer of the cornea – conjunctival infection by viruses is often also associated with infection of the cornea (keratoconjunctivitis). In the United States the direct and indirect costs of conjunctivitis have been estimated to approach 1 billion USD annually [6], with human adenoviruses (HAdVs) the cause of ~60% of all cases [7]. HAdVs are divided into 7 species (A-G), with HAdV-D comprising more than 70% of the 116 genotypes currently in Gen-Bank. HAdVs cause infections at all mucosal sites, with ocular infections particularly common. Most adenovirus infections, including those of the ocular surface, are self-limiting and resolve without serious sequela. However, in epidemic keratoconjunctivitis (EKC), affected individuals develop stromal keratitis, manifest as delayed-onset corneal subepithelial infiltrates (SEI), leading to chronic and/or recurrent visual dysfunction in a significant proportion of cases [8]. Likely due to evolution of new adenoviruses [9], EKC has recently become more frequent, more severe, and more wide spread across both Asia [10–12], and Europe [13], and viruses within HAdV-D are the most frequent cause. To date, the mechanism of SEI formation in the cornea after EKC has remained elusive [14].

HMGB1 is a multi-functional protein that acts both as a chromatin-binding protein [15] and as a danger-associated molecular pattern (DAMP) when released [16], with immune stimulating properties. Disappointingly, many foundational studies on how its oxidation state impacts its immunomodulatory functions have been either retracted or underscored with expressions of concern [17]. However, it is believed that the oxidation status of HMGB1 determines its specific role in pathogenesis [18–20]. There are three main forms of HMGB1, all-thiol or reduced HMGB1, disulfide or partially oxidized HMGB1, and fully oxidized HMGB1. The all-thiol form (reduced cysteines 23, 45 and 106) directly stimulates leukocyte recruitment via the CXCL12/CXCR4 pathway [21–23], while di-sulfide HMGB1 (partially oxidized at cysteines 23 and 45) is a potent proinflammatory cytokine with high affinity to TLR4 [24]. Prior studies using HAdV-C5 showed that adenovirus protein VII of species C can bind to and sequester HMGB1 in the infected cell nucleus, leading to suppression of immune signaling by the infected cell [25,26]. This has not been shown for viruses within HAdV-D, of for the specific cell types that HAdV-Ds infect.

Extracellular HMGB1 was first reported as expressed by macrophages treated with LPS [27]. It is now known that HMGB1 secretion can be induced by infection as well as by endogenous stimuli [28]. Because HMGB1 lacks a leader sequence, it cannot be secreted by a conventional ER-Golgi secretory pathway [29], but instead is released via cytoplasmic secretory vesicles [30]. During active secretion, acetylated HMGB1 binds to CRM1 (Chromosomal Maintenance 1, also known as Exportin 1, or XPO1) for nuclear export, and accumulates within cytoplasmic vesicles, such as lysosomes, before extracellular secretion [31]. Release of HMGB1 can also occur during cell death, including necrosis, necroptosis, apoptosis, pyroptosis, and autophagy-dependent cell death [18]. Translocation of HMGB1 from the nucleus to the cytoplasm to the extracellular space are all necessary steps in its release. Regardless of whether the release of HMGB1 occurs due to cell death, it is the specific post translational state of HMGB1 protein that determines its biological activity.

Immune cell infiltration is a well characterized and highly conserved response to tissue insult. A vast literature is focused on the processes and mediators of immune cell trafficking during and after any injury, including adhesion molecule engagement [32], cytokine and chemokine receptor-based transduction [33], and intracellular signaling [34]. However, the mechanisms of leukocyte infiltration in tissues with unique architectures can be highly specific. Compiled reviews on signaling pathways activated by injury and infection have tended to utilize cell monolayers, and have highlighted the molecular crosstalk between pathways [35–38] within the same cell type. Yet, Rowland and coworkers have demonstrated that no protein has the precisely identical function in different tissues, suggesting that mechanistic studies need also to consider the tissue and multi-cellular context of any injury response [39]. Whereas the mechanisms identified in infected cell monolayers provide important clues to molecular pathogenesis of infections, immune response mechanisms in the context of a complete tissue architecture need greater study. The cornea has three principal layers. The external surface is comprised of a stratified squamous epithelium (~50 microns thick). The internal most surface is a monolayer of endothelial cells. These two solely cellular layers bound the corneal stroma (~500 microns centrally),which comprises ~90% of the corneal thickness, and consists of a collagenous extracellular matrix populated principally by keratocytes, which when cultured in the presence of serum, are characterized as human corneal fibroblasts (HCF). Herein, we have explored the origin of a highly specific manifestation of adenovirus infection in the human cornea, including potential crosstalk between tissue-specific epithelial cells and fibroblasts. The cornea is uniquely avascular and transparent, and as such provides an exceptional model for molecular studies of infection. This is the first report showing secretion of biologically active HMGB1 by infected corneal cells that is specific both to the viral agent and the infected cell type. Exogenous HMGB1 also induces a robust expression of proinflammatory mediators by corneal stromal cells, consistent with both cell types contributing to SEI formation upon adenovirus infection of the human cornea.

## Results

### HMGB1 is released by adenovirus infected cells in cell-specific fashion

HMGB1 is an alarmin expressed during infection [40,41], an intermediate in innate immune responses to bacterial DNA [42,43], and a marker of severe SARS-CoV-2 infection [44,45]. In a mouse model of HAdV-B7 pneumonia, in which infection leads to cell death but the virus does not replicate, application of a neutralizing antibody to HMGB1 reduced

pulmonary inflammation [46]. In the cornea, HMGB1 released by dying corneal epithelial cells was previously shown to induce reparative gene expression by corneal stromal fibroblasts (human corneal fibroblasts: HCF) [47]. Keratocytes are derived from neural crest and synthesize the corneal stromal extracellular matrix [48,49]. Although infection of corneal stromal cells was suggested in experimental models to be sufficient to induce SEI formation in EKC [50], to date there is no evidence in the intact cornea of adenoviruses reaching the stromal layer to directly infect HCF.

We sought to determine if epithelial cell-derived HMGB1 contributes to the delayed-onset corneal stromal inflammation characteristic of adenoviral keratoconjunctivitis. Because the trafficking of HMGB1 between the cell nucleus and the cytoplasm and then into the extracellular space is tightly controlled, we immunoblotted for HMGB1 in each of these fractions in the two major cell types present in the anterior cornea, corneal epithelial cells, which are the first corneal cells to encounter the virus in natural infection, and HCF, the latter previously proposed as a primary source of proinflammatory mediators leading to keratitis in EKC [51]. In both tert-immortalized human corneal epithelial (THE) cells and primary human corneal epithelial cells (PCEC) infected with the highly virulent EKC virus HAdV-D37 [52], HMGB1 in cell nuclei translocated to the cytoplasm by 24 h post-infection (hpi), and then to cell supernatants (Fig 1A). The migration of HMGB1 from the cell nucleus to cytoplasm to culture supernatant was similar between THE and PCEC. HMGB1 was not expressed in the supernatants of infected primary HCF, the human lung adenocarcinoma cell line A549, or the human embryonic kidney cell line HEK293. Densitometry of the HMGB1 band on Western blots confirmed HMGB1 translocation from the nucleus to the cytoplasm to the extracellular space in a time dependent fashion upon infection, seen only in THE and PCEC (Fig 1B and 1C). No translocation of HMGB1 was evident in primary HCF or in the two noncorneal epithelial cell lines. Infection in THE cells did not alter mRNA expression for HMGB1 over 48 hpi (S1A Fig), indicating that release of HMGB1 from infected corneal epithelial cells was due to either cell death or to post-translational modifications [53] rather than to increased gene expression. Expression of viral E1A gene was evident in infected THE cells by 2 hpi (S1B Fig), and of the viral late protein pIIIa in all five cell lines by 24 hpi (S1C Fig). These latter data show that all cell types used in the study were successfully infected, and confirms cell type-specific trafficking of HMGB1 upon adenovirus infection (Fig 1D).

## Time-dependent subcellular location of HMGB1 in adenovirus infection

The results above indicate a dynamic process for HMGB1 secretion by corneal epithelial cells upon infection with adenovirus. We next sought to illuminate the subcellular mechanisms of HMGB1 translocation. In the living cell, HMGB1 resides in the nucleus, and is subsequently secreted into the extracellular milieu by first translocating to the cytoplasm, followed by trafficking through vesicles [54,55] to be released either by membrane fusion or via membrane pores [56,57]. HMGB1 translocation from the nucleus to the cytoplasm is directed by CRM1, which has been shown to bind HMGB1 for nucleus to cytoplasmic transport [58]. To better elucidate the timeline of HMGB1 secretion upon HAdV-D37 infection, THE cells were infected for various time points and cytoplasmic HMGB1 was quantified using high content microscopy (HCM). In these images, the primary object (nucleus-blue mask) was identified using three different algorithms (isodata, fixed, and triangle). A region of interest (ROI) of 20 µm was set around each nucleus (grey mask). Within each ROI region of interest, a yellow line indicates the localization of HMGB1 in the cytoplasmic region and a light gray line encircling the nucleus signifies HMGB1 within the nucleus. The movement of HMGB1 from the nucleus to the cytoplasm was time-dependent, and began at 8 hpi (Fig 2A, inset iv), with even greater release at 10 and 12 hpi (Fig 2A, insets vi and viii). At 24 hpi, a steep drop in cellular HMGB1 was noted in the majority of infected cells (Fig 2A, inset x). Quantification of HCM images was performed with approximately 60,000 cells/condition, and repeated for a total of three separate experiments. As shown in Fig 2B, HMGB1 in the cytoplasm increased steadily up to 12 hpi, and then dropped steeply by 24 hpi. These data correlate with the Western blot data (Fig 1A) showing HMGB1 translocation from the cell nucleus to the cytoplasm, and then to the supernatant over the first 24 hpi. Analysis of relative fluorescence for cytoplasmic HMGB1 at each time point after infection as compared to mock infection was consistent with the movement of HMGB1 into the cytoplasm and then into the cell supernatant (Fig 2C). By HCM, levels of HMGB1 increased only in the cytoplasm of infected cells (S2A Fig).

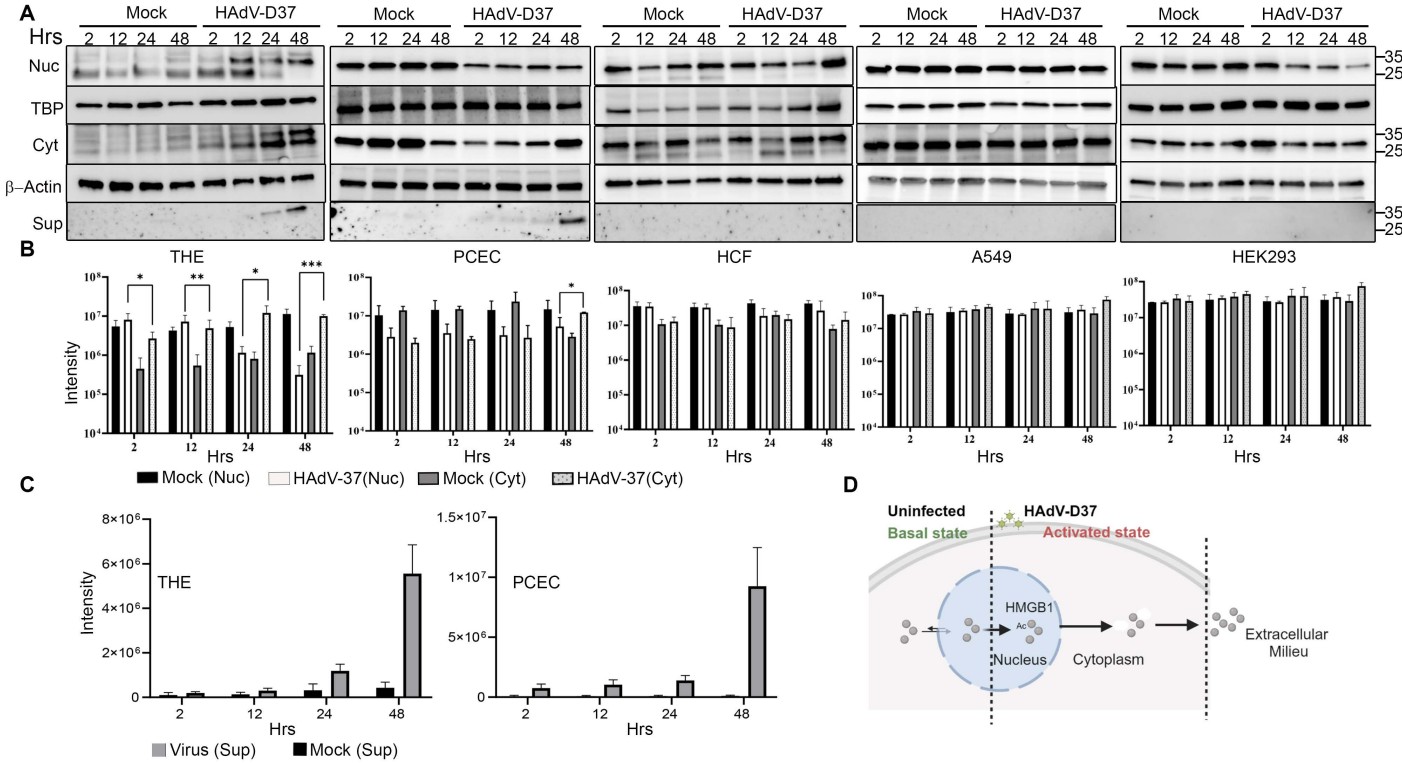

**Fig 1. Cell specific HMGB1 secretion. (A)** Cytoplasmic and nuclear extracts prepared from uninfected or HAdV-D37-infected for 2, 12, 24, and 48 hpi. were resolved on 4-20% SDS-PAGE. Western blots for HMGB1 in the nuclear (Nuc), cytoplasmic (Cyt), and supernatant (Sup) extracts with TBP (nuclear) β-actin (cytoplasmic) as loading controls in cell types THE, PHCE, HCF, A549, and HEK293 (left to right). **(B)** Densitometric analysis of HMGB1 band intensity of cytoplasmic and nuclear extracts in THE, PHCE, HCF, A549, and HEK293 (left to right). p values were determined by t test; *$p < 0.05$, **$p < 0.01$, ***$p < 0.001$, ****$p < 0.0001$ ($n = 3$). **(C)** Quantification of extracellular HMGB1 measured from supernatants collected at indicated time points pi in THE and PHCE cells ($n = 3$). **(D)** Schematic representation (created using Biorender.com) for HAdV-D37 induced translocation of HMGB1 from the cell nucleus to the cytoplasm and then into the extracellular space.

A comparison between HMGB1 fluorescence in the nuclei vs. the cytoplasm within the infected cells confirmed a gradual increase in nucleus-to-cytoplasmic translocation of HMGB1(S2B Fig).

## HMGB1 translocation is specific to HAdV species

Following the observation of cell specific HMGB1 translocation after HAdV-D37 infection, we tested several different HAdVs to explore whether HMGB1 secretion by corneal epithelial cells was also virus-specific. HAdV-C5 is a well-studied adenovirus type important to studies of viral oncogenesis [59] and the backbone of a major SARS-CoV-2 vaccine [60]. In A549 cells, HAdV-C5 protein VII was shown to bind HMGB1 to repress innate immune responses and promote infection [25,26]. HAdV-D56 was identified and associated with several fatalities in a neonatal-intensive care unit and keratocon-junctivitis in three caregivers [61]. HAdV-D9 has not been previously identified as a cause of EKC, but exhibits corneal tropism *in vitro* [62]. At the relatively low multiplicity of infection (MOI) of 1 and at 24 hpi, cytopathic effect was induced in THE cells infected with HAdV-D37, 56, and 9 and to a lesser extent with HAdV-C5 (Fig 3A). All 4 viruses caused cyto-pathic effect in HCF under the same conditions. HMGB1 translocation to the cytoplasm was noted only in THE cells infected with cornea-tropic HAdV-D (Fig 3B), not with HAdV-C5 (Fig 3B). Notably, HMGB1 cytoplasmic translocation also occurred with infection by HAdV-D9 (Fig 3B), earlier reported to have corneal epithelial cell tropism due to evolutionary

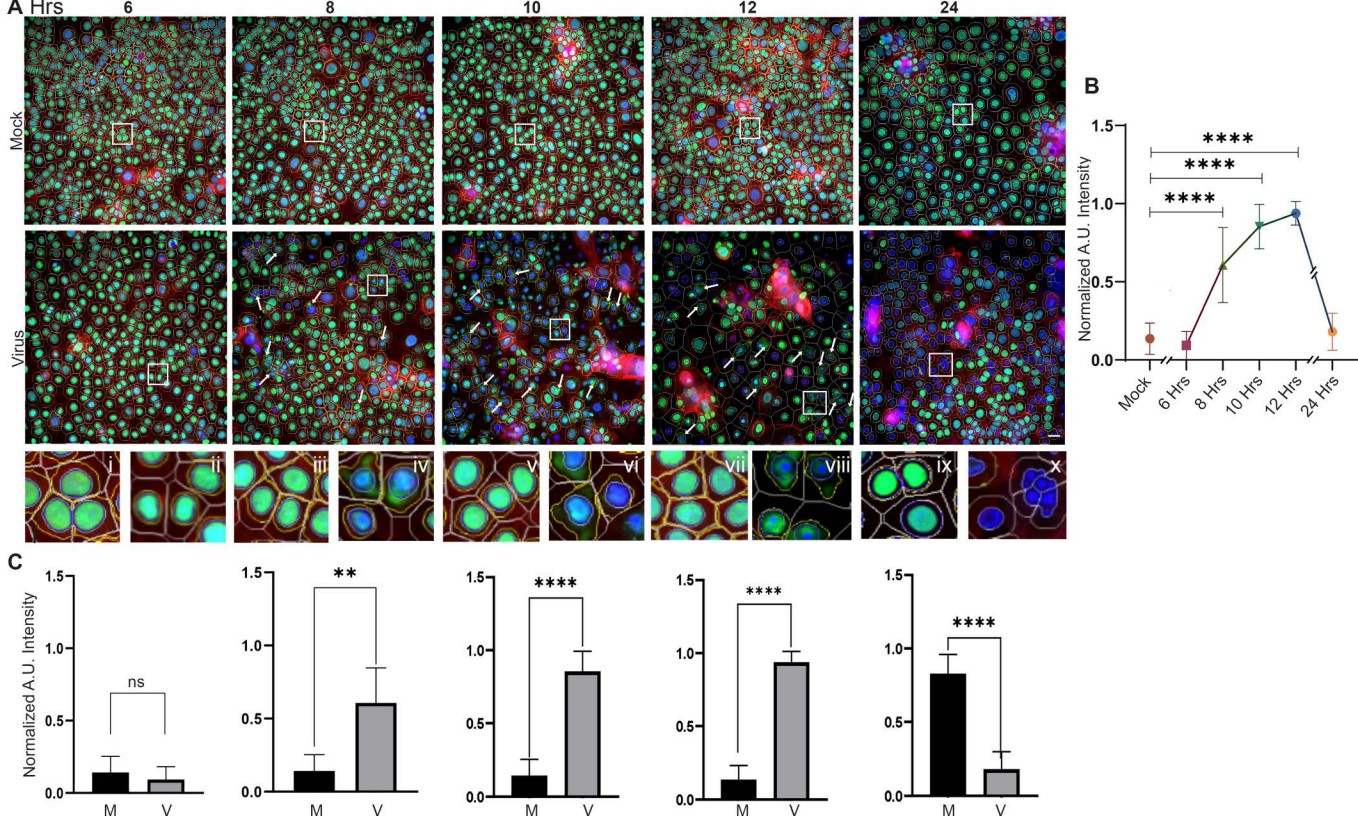

**Fig 2. Time dependent translocation of HMGB1 in HAdV-37 infection. (A)** Nuclear and cytoplasmic HMGB1 (green) at 6, 8, 10, and 12 hpi was quantified using computer generated region of interest (ROI) analysis around the nucleus and cell boundary. Arrows and insets (i to x) show HMGB1 nucleus-to-cytoplasmic translocation starting at 8 hpi. At 24 hpi, most of the HMGB1 signal is gone from the cytoplasm, due to secretion into the extra-cellular space ($n = 5$). **(B)** Quantitative analysis of cytoplasmic HMGB1 shows a steady increase until 12 hpi, then a steep drop at 24 hpi. p values at 6 (0.959, ns), 8 (<0.0001), 10 (<0.0001), 12 (<0.0001), 24 hpi (0.9265) compared to mock infection. p values were calculated by one-way ANOVA (Tukey's multiple comparisons test). **(C)** Graphs of cytoplasmic HMGB1 measured at various time points pi and compared to mock infection show increases at 8 (<0.0033), 10 (<0.0001), and 12 hpi (<0.0001), and a decrease at 24 hpi (<0.0001). Statistical testing was performed by unpaired t-test (two-tailed). Analysis was done on 60,000 cells/group ($n = 5$).

pressure on a single amino acid at position 240 in the fiber knob [62]. Immunoblotting of supernatants and cytoplasmic lysates for THE and HCF at 24 hpi showed HMGB1 only in the supernatants of THE cells and only with viruses from HAdV-D (Fig 3C). All four viruses infected both cell types based on the expression of late viral proteins when immunoblotted with a pan-HAdV-C5 antibody (Fig 3C). Phylogenetic analysis between the adenoviruses used in these experiments showed the close relationship between the whole genomes of HAdV-D9, 37, and 56 (Fig 3D), with branch lengths indicating a relatively small degree of genetic change. An earlier study with HAdV-C showed HMGB1 binding to protein VII in the infected cell nucleus, resulting in sequestration in the nucleus [26]. To determine if the previously reported interaction between protein VII and HMGB1 is specific to HAdV-C, we investigated protein VII conservation across representative viruses from both C and D HAdVs. The protein VII amino acid identity and similarity scores comparing HAdV-D (types 9, 37, and 56) and HAdV-C (types 2 and 5) ranged between 72.41 to 73.63% and 79.31 to 80.60%, respectively (Fig 3E). Western blot analysis comparing nuclear extracts from HAdV-D37 and HAdV-C5 infected THE cells showed protein VI but not protein VII in the nuclei of HAdV-D37 infected cells. In contrast, both proteins were present in HAdV-C5 infected nuclear extracts at 24 and 48 hpi (Fig 3F). Taken together, these data suggest that protein VII of HAdV-D does not

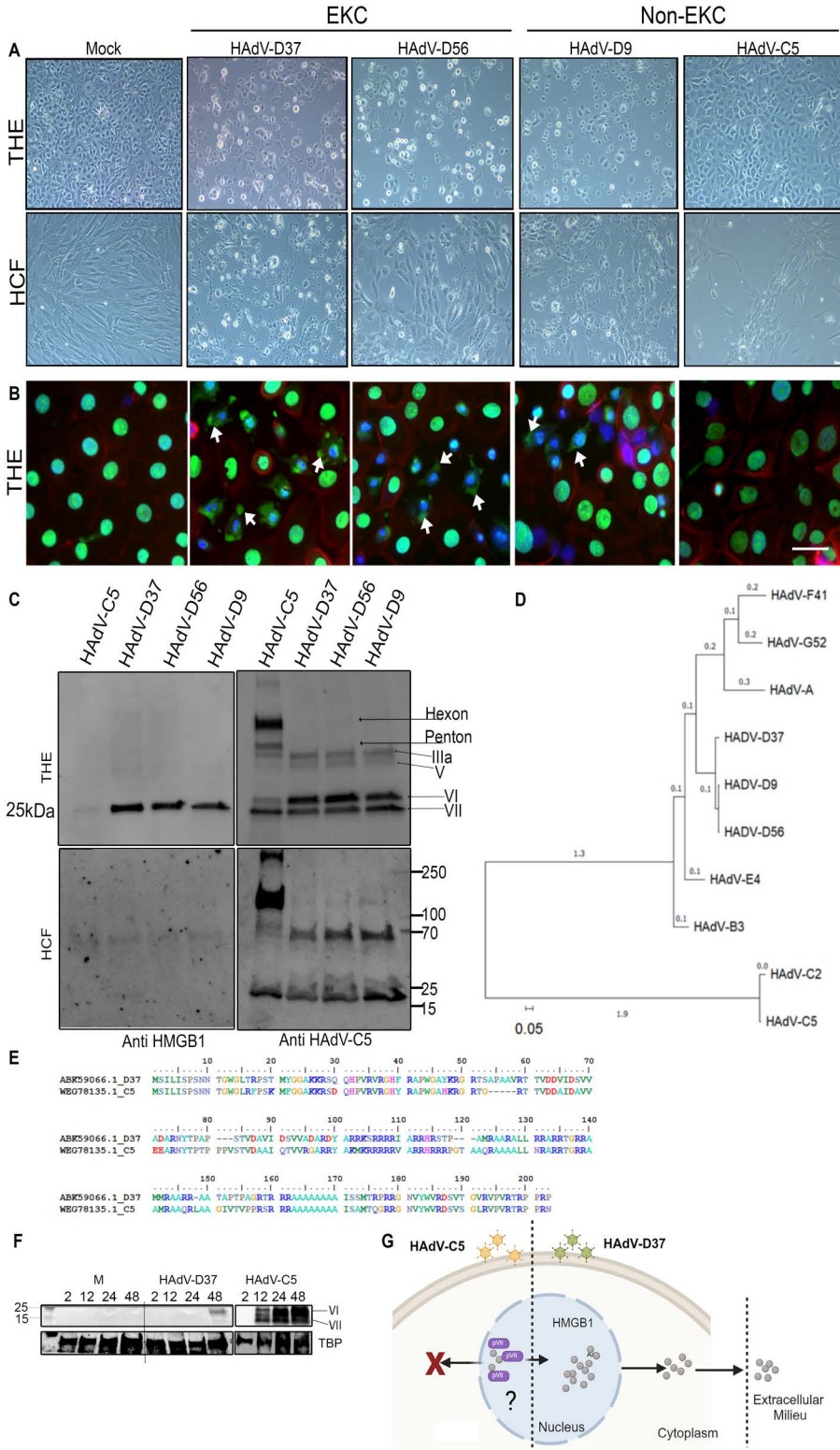

**Fig 3. EKC virus-specific HMGB1 secretion. (A)** Light microscopic images show cytopathic effect induced by HAdV-D37, D56, D9, and C5 infection of THE and HCF compared to uninfected cells at 24 hpi. Cytopathic effect was noted in all HAdV-D infections, and to a lesser degree with HAdV-C5 infection. **(B)** Immunofluorescence images at 24 hpi show HMGB1 (green) translocation to the cytoplasm (white arrow heads) only with HAdV-D37, D56, and

D9 infection, but not in C5 infection (Space bar = 10μm) *(n = 3)*. **(C)** Cell culture supernatants from THE and HCF infected with, HAdV-D37, D56, D9, and C5, or mock treated cells at 24 hpi were probed with anti-HMGB1 antibody. Western blot from cellular extracts from the same experiment probed with HAdV-C5 pan antibody are shown in in the adjacent panel. **(D)** Phylogenetic analysis of whole adenovirus genomes. There were a total of 38,913 positions in the final data set. The evolutionary history was inferred using the Maximum Likelihood method and Tamura-Nei model [105]. Initial tree(s) for the heuristic search were obtained by applying Neighbor-Join and BioNJ algorithms. Evolutionary analyses were conducted in MEGA11 [106]. **(E)** Multiple sequence alignment of representative protein VII amino acids for members of HAdV-C and D using BioEdit. Sequence differences across types are color coded and sequence conservation are shown with dots. **(F)** Nuclear extracts from HAdV-D37 and HAdV-C5 infected cells were probed with adenovirus pan antibody for expression of protein VI and protein VII as indicated. **(G)** Schematic representation (created using Biorender.com) of virus-specific HMGB1 secretion comparing HAdV-C5 and D37.

sequester HMGB1 in the nucleus, as reported for HAdV-C, and indicate a distinctly different role for protein VII in HMGB1 biology between these two HAdV-species (Fig 3G).

## HMGB1 acetylation is cell and virus specific

Acetylation of HMGB1 protein at its nuclear localization sequences (NLS) 1 and 2 leads to nuclear-cytoplasmic shuttling [18,55], while hyperacetylation prevents nuclear return, and leads to accumulation of HMGB1 in the cytoplasm and eventually trafficking through secretory vesicles [55]. Acetylation is therefore a pre-requisite to HMGB1 secretion for subsequent proinflammatory signaling. As HAdV-Ds with corneal epithelial tropism specifically induce HMGB1 translocation and secretion, we sought to determine if HMGB1 acetylation was also specific to cell type. By confocal microscopy, the prototype EKC virus HAdV-D37 induced HMGB1 acetylation in corneal epithelial cells (THE and PCEC), but not in HCF or A549 cells (Fig 4A). In contrast, HAdV-C5 did not induce HMGB1 acetylation in any of these cell types under the experimental conditions (Fig 4A), supporting both cell and virus specific HMGB1 acetylation. To determine whether HMGB1 acetylation and release was an active process or a bystander effect due to cytolysis, THE cells were infected and assayed for 48 hpi. Cytopathic effect was demonstrable only after 24 hpi (Fig 4B). In contrast, immunoblot performed on cell supernatants showed acetylated HMGB1 in the cell culture media at just 8 hpi (Fig 4C), well before the onset of cytopathic effect. Quantification of HCM images for acetylated HMGB1 in infected vs. mock-infected THE cells using HCM showed greater acetylated HMGB1 in the cytoplasm of infected cells (Fig 4D). These results indicate virus-specific HMGB1 secretion is also cell-specific (Fig 4E), and that extracellular HMGB1 expression by infected corneal epithelial cells is an active process that occurs prior to cell death.

## HMGB1 secretion upon viral infection is independent of pyroptosis

Cell death invariably results in some release of cellular contents into the extracellular milieu, regardless of the death pathway [63–67]. Whether release of HMGB1 specific to HAdV-D37 infection of corneal epithelial cells is a bystander effect of cell death, or a specific cellular pathway unrelated to cell death is unclear. In apoptotic cell death, hypoacetylated HMGB1 remains tightly bound to chromatin and is functionally mute [65]. HMGB1 release through pyroptosis is potently proinflammatory. In immune cells undergoing pyroptosis, cell membrane leakage leads to release of hyperacetylated, active HMGB1 [68], along with IL-β and IL-18 [69] (Fig 5A). Pyroptosis has been demonstrated previously in adenovirus infection of leukocytes [70,71], and in human corneal epithelial cells infected with herpes simplex virus [72] or exposed to airborne particulates [73], but has not been studied in adenovirus-infected human corneal epithelial cells. HAdV-D37 infection of THE cells at an MOI of 1 leads to cytopathic effect with visible cell death beginning at ~ 24 hpi (Fig 4B). The appearance of HMGB1 in cell supernatants over time, as compared to the time course of cytopathic effect seen in corneal epithelial cells infected with HAdV-D37, is consistent with HMGB1 secretion prior to viral cytopathic effect. To discern whether HMGB1 secretion following adenovirus infection of corneal epithelial cells might occur through a pyroptotic pathway, we tested for byproducts of pyroptosis up to 48 hpi. By immunoblots on supernatants of infected THE and PCEC, there was no release of either IL-1β or IL-18 (Fig 5B). GAPDH expression was seen only in the infected PCEC supernatants and by

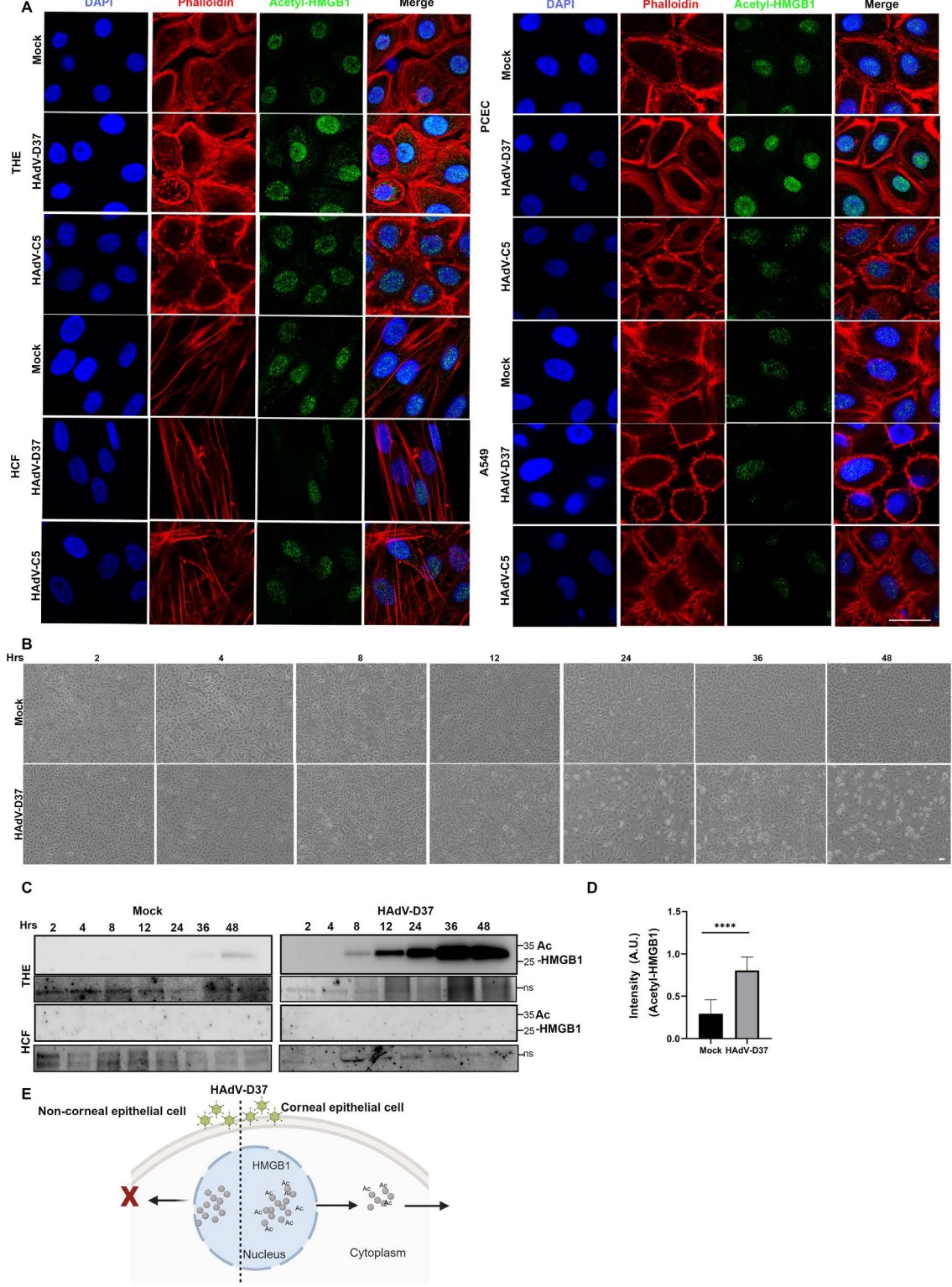

**Fig 4. HMGB1 acetylation is virus specific. (A)** Confocal microscopy using anti-acetylated HMGB1 antibody at 24 hpi with HAdV-D37 or C5, and in 4 cell types: THE, HCF, PCEC and A549. Columns from left to right: nuclei (blue), actin (red), acetyl-HMGB1 (green), and overlay. Increased acetylated-HMGB1 was observed only in THE and PCEC. Scale bar = 10μ. **(B)** Light microscopy images showing cytopathic effect of HAdV-D37 infected THE cells compared to uninfected cells at different time points (*n = 3*). **(C)** Western Blot performed on supernatants collected and concentrated to 50%

volume. Acetylated HMGB1 was evident beginning from 8 hpi up to 48 hpi, but not observed in concentrated supernatants of infected HCF. For load control, a nonspecific band (ns) is shown below each panel. **(D)** Quantitative analysis of HCM images shows increased acetylated-HMGB1 in the cytoplasm of HAdV-D37 infected THE cells compared to mock treated cells. Cytoplasmic HMGB1 quantification from 60,000 cells/condition (MOI of 5) (n = 3). **(E)** Schematic (created using Biorender.com) of acetylated HMGB1 trafficking upon specific cell type infection.

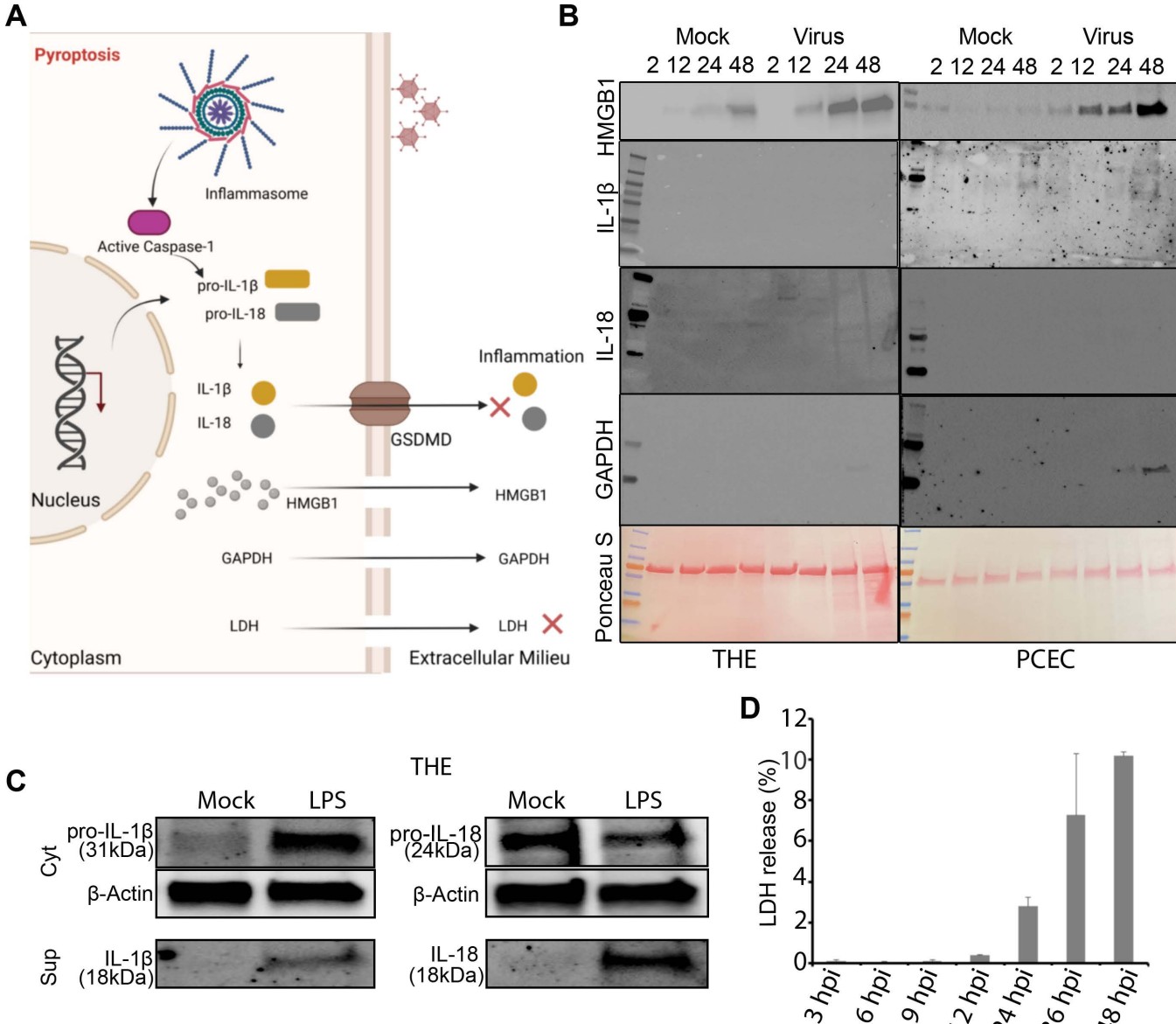

**Fig 5. Corneal epithelial cells release of HMGB1 is independent of pyroptosis. (A)** Schematic representation (created using Biorender.com) of corneal epithelial cell release of HMGB1. HAdV-D37 induces release of HMGB1 into the extracellular space but not IL-18 or IL-1β. **(B)** Cell culture supernatants from HAdV-D37 infected THE and PCEC, or mock infected cells, were probed with antibodies to HMGB1, GAPDH, IL-18, and IL-1β. Only HMGB1 was detected in the supernatants of THE and PCEC infected cells. Load control and protein presence is indicated by ponceau S staining of the blots. In contrast, inflammasome activity was induced by LPS treatment, as evident by the expression of IL-18 and IL-1β (n = 3). **(C)** Western blots for IL-18 and IL-1β induced by LPS treatment of THE cells. **(D)** Colorimetric assay for LDH in the supernatants of HAdV-D37 infected THE cells. LDH quantity was below cytotoxicity levels even at 48 hpi. Data with error bars are represented as mean ± SD.

48 hpi (Fig 5B) was still minimal. To check the efficiency of the antibody, we also performed Western blot analysis of cell lysates and supernatants after infection and on LPS treated cells, using the same cell lines (Fig 5C). Furthermore, while lactate dehydrogenase activity (LDH) in infected THE cells appeared to increase over time, it remained below biologically significant levels for the degree of cytopathic effect, even at 48 hpi (Fig 5D). When infected cell supernatants were examined with a comprehensive cytokine array, neither THE cell supernatants (S3A and S3B Fig) nor PCEC supernatants (S4A and S4B Fig) showed an expression pattern consistent with pyroptosis, further confirming that virus-infected cells release HMGB1 in the absence of pyroptosis.

## CRM1 dependent HMGB1 translocation upon adenoviral infection

To determine the molecular pathway of HMGB1 secretion in corneal epithelial cells, we further examined HMGB1 intracellular trafficking upon infection. HMGB1 translocation from the nucleus to the cytoplasm is tightly regulated by CRM1 [31,74]. We applied HCM using CRM1 knockdown (siCRM1) and negative control (NC-siRNA) treated cells followed by adenovirus infection for 12 h, a time when HMGB1 nuclear to cytoplasmic translocation was clearly evident in prior experiments (Fig 2A). As shown (Fig 6A), HMGB1 remains in the nucleus in the absence of infection, but has translocated to the cytoplasm by 12 hpi (Fig 6A, inset ii). In contrast, when cells were pretreated with CRM1 specific siRNA, HMGB1 remained sequestered in the nucleus of infected cells (Fig 6A, inset iv). Quantitative analysis of ~ 60,000 cells/condition in three separate experiments confirmed this (Fig 6B). siCRM1 pretreated, infected cells had similar levels of nuclear HMGB1 as in NC-siRNA pretreated, uninfected cells, and also similar levels of cytoplasmic HMGB1. siCRM1 pretreatment significantly increased nuclear HMGB1 ($p < 0.0015$) and reduced cytoplasmic HMGB1 ($p < 0.0001$) in infected cells. These results indicate that HMGB1 translocation to the cytoplasm in adenovirus infection is specifically mediated by CRM1. Western blots from parallel experiments confirmed successful CRM1 knockdown, and less HMGB1 in supernatants of CRM1 knocked down, infected cells (S5A Fig). Confocal microscopy was also consistent with co-localization of HMGB1 and CRM1 in the nucleus upon HAdV-D37 infection, as compared to mock controls (S5B Fig).

## HMGB1 trafficking through the cytoplasm

The mode of HMGB1 expression in the extracellular milieu has been much debated [28,30,75] and may differ by cell type and depend on the insult. We further sought to understand the cytoplasmic trafficking of HMGB1 in adenovirus infected corneal epithelial cells. HMGB1 was previously shown to partially colocalize in LAMP1 containing vesicles [30]. Pretreatment with LAMP1 siRNA at 12 hpi had no effect on translocation from the nucleus to the cytoplasm (Fig 6C, inset ii and D), but in siLAMP1 pretreated cells at 24 hpi, a time when we previously noted HMGB1 had been released from infected cells (Fig 2A), HMGB1 levels in both the nucleus and cytoplasm were greater than in NC-siRNA pretreated cells (Fig 6E, inset iv and F, $p < 0.025$ and $p < 0.0015$ respectively). Western blot showed less HMGB1 in the culture supernatants of siLAMP1 treated cells at 24 hpi as compared to control NC-siRNA treated cells (S5C Fig). By confocal microscopy, colocalization of HMGB1 and LAMP1 was evident in the cytoplasm of virus infected cells with greater nuclear to cytoplasmic HMGB1 localization as compared to mock infected cells (S5D and S5E Fig). A model for the putative contributions of CRM1 and LAMP1 on extracellular trafficking of HMGB1 in adenovirus infection is shown in Fig 6G.

## rHMGB1 induces proinflammatory cytokine expression by corneal stromal cells

The above data show that HMGB1 is secreted by corneal epithelial cells upon adenovirus infection in a tightly regulated fashion. The corneal epithelium is situated at the surface of the corneal stroma, the latter populated by keratocytes. To plumb the effect of epithelial expression of HMGB1 on underlying stromal cells, we treated primary HCF with rHMGB1 or buffer control, and compared the proteomes using a cytokine array. As shown (Fig 7A and 7B), multiple proinflammatory cytokines were upregulated as compared to uninfected, buffer treated cells. Of note, CXCL12 was upregulated in

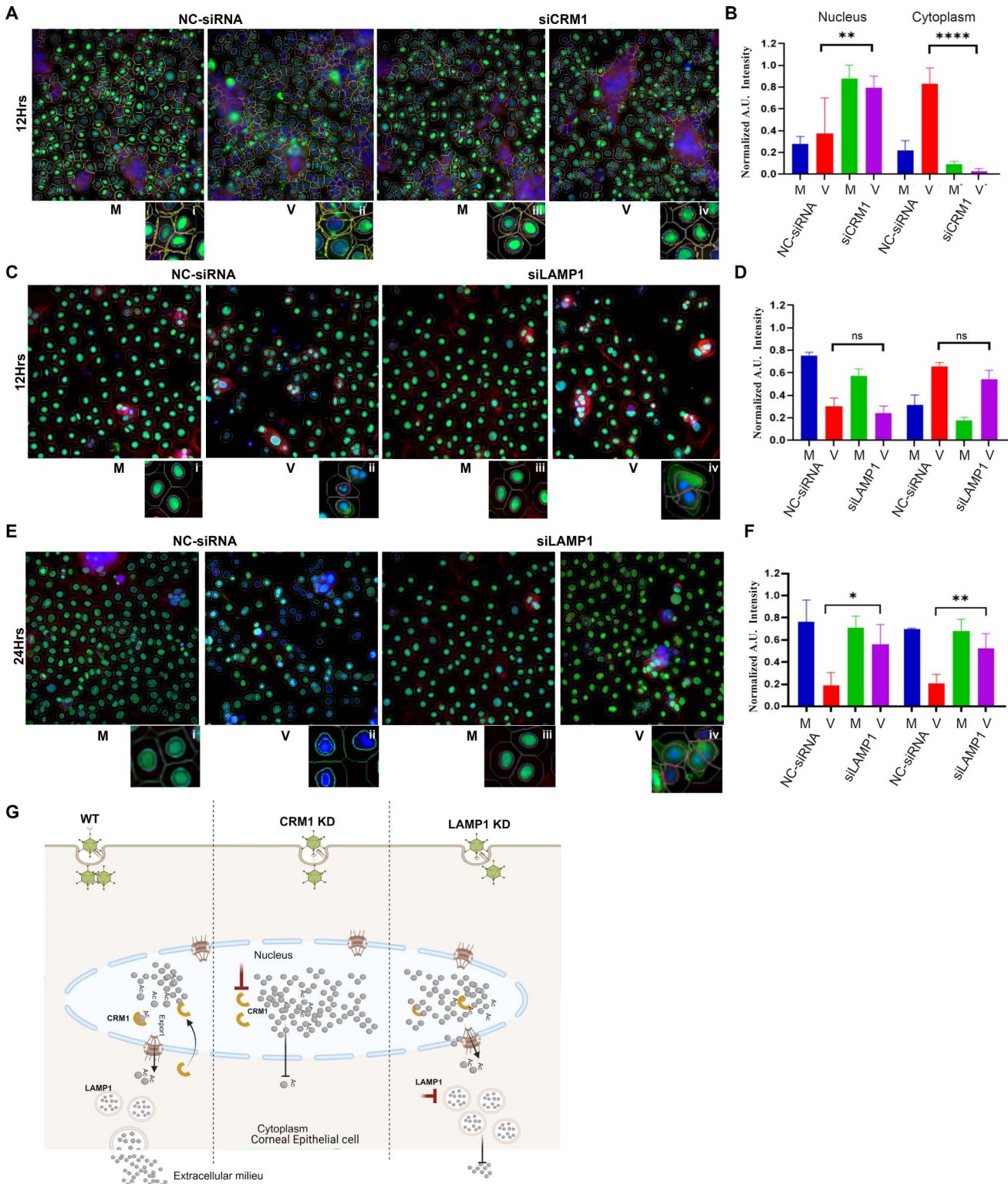

**Fig 6. CRM1 dependent HMGB1 translocation and trafficking to the extracellular space upon adenoviral infection. (A)** CRM1 knockdown (siCRM1) or control (NC-siRNA) treated THE cells, infected with HAdV-D37 for 12 h for HCM analysis. HMGB1 translocation to the cytoplasm is observed in infected, NC-siRNA pretreated cells (inset ii, yellow arrow). No HMGB1 cytoplasmic translocation was observed in siCRM1 treated, or in mock infected cells (inset i, iii, and iv). Masks: gray, algorithm-defined cell boundaries; blue, computer-identified nucleus; yellow outline,

computer-identified cytoplasmic HMGB1. Scale bar = 10 μm. **(B)** Quantitative analysis shows increased nuclear HMGB1 retention in siCRM1 treated, virus infected cells, as compared to NC-siRNA treated, infected cells (red vs purple bars). Cytoplasmic quantification shows reduced HMGB1 in siCRM1 treated, infected cells as compared to NC-siRNA treated, infected cells (red vs purple bars) ($n = 5$). **(C)** LAMP1 knockdown (siLAMP1) or control (NC-siRNA) treated cells infected with HAdV-D37 for 12 h. HMGB1 translocation is seen in both NC-siRNA treated and siLAMP1 treated, infected cells (inset ii and iv, yellow arrows) as compared to mock infected cells (inset a and c) **(D)** HCM quantification shows no significant change in nuclear HMGB1 or cytoplasmic translocation in either NC-siRNA or siLAMP1 treated, infected cells, or mock infected cells ($n = 3$). **(E)** Confocal analysis at 24 h after LAMP1 siRNA treatment. In NC-siRNA treated and infected cells no HMGB1 is seen in the cytoplasm, similar to Fig 1A (inset ii, yellow arrow). siLAMP1 treatment led to cytoplasmic HMGB1 retention at 24 hpi (inset iv, yellow arrow). **(F)** Similar HCM analysis at 24 hpi showed modest nuclear HMGB1 retention in siLAMP1 treated, virus infected cells, as compared to NC-siRNA treated, infected cells (red vs purple bars). In the cytoplasm, HMGB1 accumulation was noted in siLAMP1 treated, virus infected cells as compared to NC-siRNA treated, infected cells. Data shown as the mean ± SD ($n = 3$). Analyzed by ANOVA with Tukey's post-hoc test. **(G)** Schematic representation (created using Biorender.com) showing the pathway of HMGB1 release upon infection of corneal epithelial cells. HMGB1 is dependent on CRM1 for nuclear to cytoplasmic shuttling, and LAMP1 for extracellular release.

rHMGB1 treated HCF. It was previously shown that HMGB1 forms a heterocomplex with CXCL12 for mononuclear cell recruitment [23].

To identify proteins of high significance, proteins were first sorted by their ascending p-value and then listed by the absolute values of logarithmic conversion. A positive value indicates upregulation, and a negative value indicates a downregulation in rHMGB1 treated HCFs relative to control buffer. Significantly up- or downregulated proteins are listed in Fig 7. The most significantly upregulated proteins identified were clustered in the group termed 'cell chemotaxis' associated with Biological Process; and cytokine and chemokine activity terms in the category of Molecular Function. Analysis using ShinyGO 0.80 indicated that in the category Biological Process, the upregulated genes were enriched in the following groups: 'cell chemotaxis' (GO:0060326), 'leukocyte chemotaxis' (GO:0030595), 'regulation of leukocyte chemotaxis' (GO:0002688), 'myeloid leukocyte migration' (GO:0097529), 'inflammatory response' (GO:0006954), 'response to external stimulus' (GO:0009605), 'chemokine-mediated signaling pathway' (GO:0070098), 'granulocyte chemotaxis' (GO:0071621), 'cell migration' (GO:0016477), 'positive regulation of leukocyte chemotaxis' (GO:0002690), 'cytokine-mediated signaling pathway' (GO:0019221), 'cellular response to cytokine stimulus' (GO:0071345), 'cellular response to organic substance' (GO:0071310), 'regulation of cell migration' (GO:0030334), 'defense response' (GO:0006952), 'neutrophil chemotaxis' (GO:0030593), 'regulation of response to external stimulus' (GO:0032101), 'regulation of anatomical structure morphogenesis' (GO:0022603), 'response to oxygen-containing compound' (GO:1901700), 'regulation of cell population proliferation' (GO:0042127), 'humoral immune response' (GO:0006959), and 'positive regulation of cell migration' (GO:0030335) (Fig 7E). In the category of Molecular Function, the upregulated genes were mainly enriched in 'cytokine activity' (GO:0005125), 'receptor ligand activity' (GO:0048018), 'chemokine activity' (GO:0008009), 'cytokine receptor binding' (GO:0005126), 'signaling receptor binding' (GO:0005102), 'CXCR chemokine receptor binding' (GO:0045236), 'growth factor activity' (GO:0008083), 'protein binding' (GO:0005515), 'molecular function regulator' (GO:0098772), 'fibronectin binding' (GO:0001968), 'extracellular matrix binding' (GO:0050840), 'CCR2 chemokine receptor binding' (GO:0031727), 'insulin-like growth factor II binding' (GO:0031995), 'insulin-like growth factor I binding' (GO:0031994), 'growth factor binding' (GO:0019838), 'integrin binding' (GO:0005178), and 'heparin binding' (GO:0008201) (Fig 7F). Furthermore, in the category Cellular Component, the significantly enriched genes were concentrated in the following terms: 'extracellular space' (GO:0005615), 'secretory granule lumen' (GO:0034774), 'platelet alpha granule lumen' (GO:0031093), and 'endoplasmic reticulum lumen' (GO:0005788) (Fig 7G). All of the aforementioned terms were significantly enriched by the differentially expressed proteins (p < 0.05). KEGG pathway analysis revealed that the upregulated proteins were significantly enriched in 'rheumatoid arthritis' (hsa05323; 7 proteins), followed by 'viral protein interaction with cytokine and cytokine receptor' (hsa04061), 'IL-17 signaling pathway' (hsa04657), 'malaria' (hsa05144), 'cytokine-cytokine receptor interaction' (hsa04060), and 'chemokine signaling pathway' (hsa04062), (Fig 7H). STRING analysis was employed to determine the roles of upregulated genes in the rHMGB1 treated HCFs along with protein-protein interaction network (Fig 7C). The network statistics were as follows: number of edges, 93; number of nodes, 18; average node

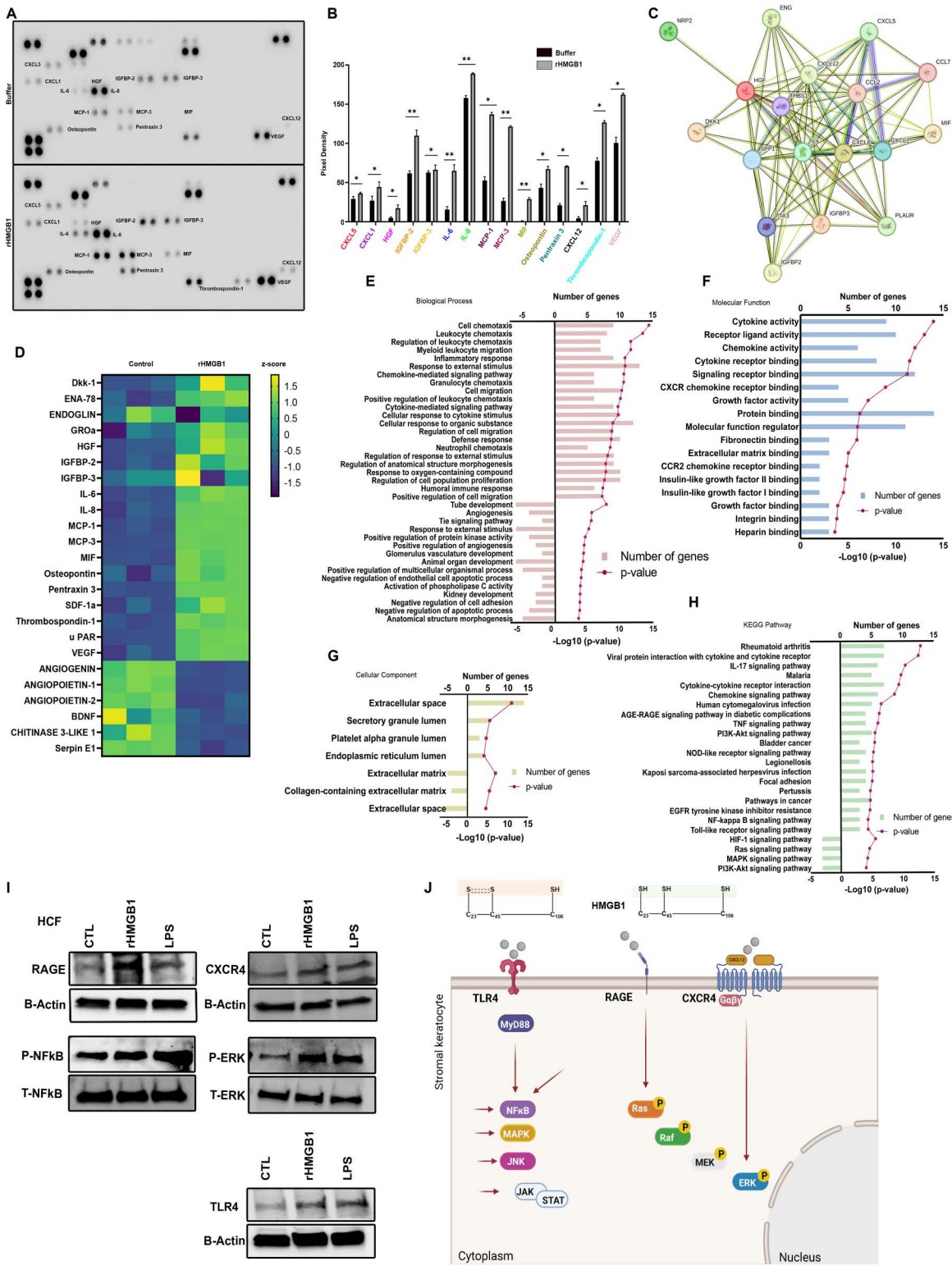

**Fig 7. rHMGb1 induces expression of proinflammatory mediators in human corneal fibroblasts.** (A) Human cytokine array blot performed on cell supernatants from HCF treated with either PBS (upper panel) or 2 µg/ml rHMGB1 (lower panel) for 4 h showing upregulation of proinflammatory cytokines. (B) Graphical representation of proteins quantified using ImageJ from three experiments, and compared between buffer and rHMGB1 treated cell supernatants. (C) The physical and functional associations among the upregulated proinflammatory mediators were assessed using the STRING tool.

The interaction among the query proteins represents the network with 18 nodes and 93 edges of protein-protein interaction (PPI). (D) Heatmap showing the differentially expressed cytokines. Heatmap rows depict z-score SD variation from the mean value for each cytokine ($n = 3$). (E-H) Gene ontology (GO) showing top biological process, molecular function, KEGG pathway, and cellular components. Bar lengths represent the number of genes and dotted line represents -log10 adjusted p value for significantly enriched pathways. (I) Effect of rHMGB1 on receptors (RAGE, CXCR4, TLR4), and the activation of NFκB and ERK. HCF were treated with rHMGB1 or LPS (100 ng/ml) for 4 h and total cell lysates were prepared to visualize RAGE, CXCR4, TLR4, p-NFκB/total NFκB and p-ERK/total-ERK expression by Western blot. β-actin was used as an internal loading control. Representative blots of three independent experiments are shown. (J) Schematic (created using Biorender.com) of predicted receptor/signaling pathways activated in HCF upon HMGB1 stimulation.

degree, 10.3; average local clustering coefficient, 0.83; PPI enrichment P value < 1.0x10-16. This level of enrichment in the rHMGB1 treated HCFs indicated significant upregulation of proinflammatory cytokines and chemokines.

Both RAGE and TLR4 act as cellular receptors for HMGB1 [76,77], and their interaction with one another promotes HMGB1-induced inflammation [78]. By Western blot, treatment of HCF with rHMGB1 induced upregulation of RAGE as compared to control buffer or LPS (Fig 7I). In HCF exposed to rHMGB1, CXCR4 and TLR4 were both upregulated (Fig 7I). Downstream signaling from RAGE/TLR4 engagement was also demonstrated by increased pERK1/2 and NFκB after either rHMGB1 or LPS treatment as compared to controls, indicating a potential pathway for HMGB1 activation of proinflammatory gene expression in keratocytes via RAGE-ERK1/2-NFκB-ERK1/2. A putative pathway for HMGB1 signaling in keratocytes is shown (Fig 7J).

To better understand the complexity of corneal epithelial cell responses to adenovirus infection beyond HMGB1 secretion, and the potential for additional cross-talk with the underlying keratocytes, we also infected or mock-infected PCEC and THE cells for proteome analysis. Earlier reports suggested that human corneal epithelial cell expression of proinflammatory mediators at early times post adenovirus infection was minor and unlikely to contribute to the formation of SEI [79]. At 24 hpi, THE (S3 Fig) and PCEC (S4 Fig) expressed a number of inflammatory mediators, overlapping to some degree with those expressed by HCF when exposed to rHMGB1 (Fig 7). Notable among those mediators expressed by both corneal epithelial cell types but not by HCF was IP10 and IL-1α. STRING, gene ontology, and KEGG pathway analyses between PCECs and THE cells were comparable. A Venn diagram of the mediators induced by all three cell types and where they overlap provides a more holistic look at the sum of inflammatory mediators induced in the cornea upon adenovirus infection (S4H Fig).

### 3D corneal culture and infection

All of the experiments described above were performed in cell monolayers, which imperfectly recapitulate corneal tissue. We next applied an optimized 3-dimensional cornea facsimile (cornea in a test tube) [80] to determine if corneal epithelial cell infection and subsequent HMGB1 expression were sufficient to induce SEI in the underlying corneal stroma. Cornea facsimiles were constructed in 3μ pore size transwell inserts (Fig 8A), with or without keratocytes, and either infected or treated with rHMGB1. Peripheral blood mononuclear cells (PBMC) were added to the bottom chamber after infection (Fig 8B). To account for possible differences between PCEC and THE cells, corneal facsimiles were constructed separately with each cell type. In facsimiles with keratocytes, by 12 hpi, PBMC had migrated against gravity, and appeared in the stroma as isolated foci just below the epithelial cell layer (Fig 8C, H&E, white arrows; DAPI stain, yellow arrows), similar in location and character to SEIs in the adenovirus infected human cornea. At 24 hpi in both PCECs and THE cells, H&E staining showed abundant immune cell migration in a dispersed pattern (white arrows) along with damage to the extracellular matrix. DAPI staining showed immune cell clustering (yellow arrows) at the subepithelial region distinguishable by cell size (S6 Fig).

FSSE is a tetramer peptide which binds to the TLR4 adaptor molecule myeloid differentiation factor 2 (MD-2) to block TLR4-dependent HMGB1 signaling [24]. In corneal facsimiles to which we added FSSE, immune cell migration into the overlying stroma was reduced (Fig 8C). Western blot analysis of the facsimiles and their supernatants also showed

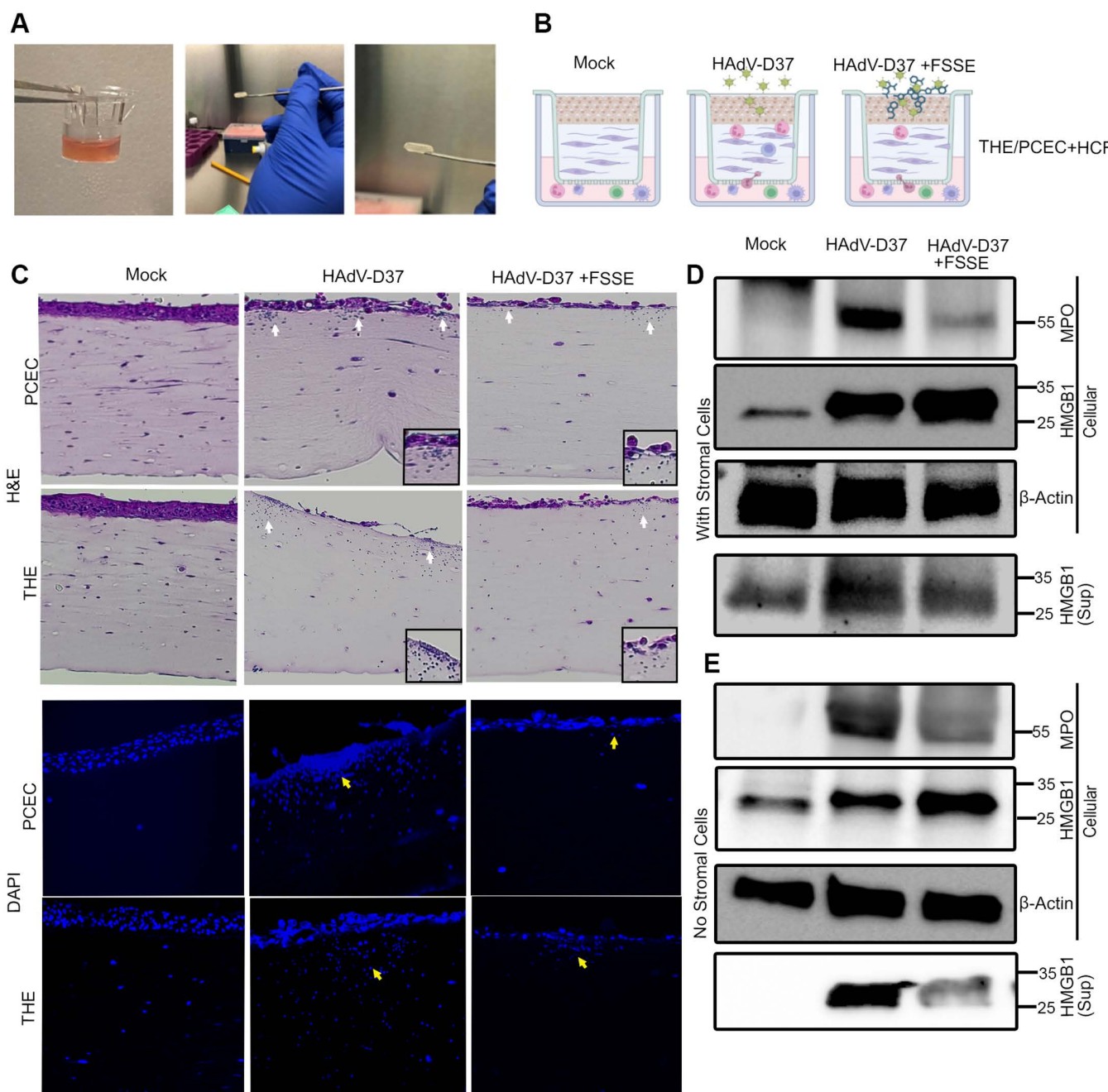

**Fig 8. 3D culture mimics natural disease.** (A) Photographs depicting 3D corneal construct preparation. (B) Schematic representation of the 3D corneal constructs treatments and experimental design. (C) Histology of the corneal construct. H&E staining of the mock infected corneal construct with PCEC and THE shows stratified epithelium and stroma with no PBMC migration (column 1, panels 1 and 2). HAdV-D37-infected corneal constructs show foci of immune cell migration as indicated by white arrows (column 2, panels 1 and 2), and reduced immune cell infiltration when pretreated with FSSE (column 3, panels 1 and 2). The enlarged insets show migratory cells within an inflammatory foci. DAPI staining shown for constructs from the same groups, and immune cell migration upon HAdV-D37 infection (column 2, panels 3 and 4, yellow arrow), with reduced immune cells in FSSE treated constructs (column 3, panels 3 and 4, yellow arrows). (D) Immunoblotting of the corneal construct with stromal cells shows increased MPO expression compared to mock infected constructs and to FSSE peptide pre-treated constructs (top panel). Total cellular HMGB1 was retained when constructs were pretreated with FSSE peptide as compared HAdV-D37 infected constructs without FSSE (panel 2). A corresponding increase in secreted HMGB1 in supernatants is seen in HAdV-D37 infected constructs compared to FSSE pretreated constructs (panel 4). Actin for loading control is shown in panel 3.

(E) In constructs with no stromal cells, the patterns of HMGB1 and MPO expression are similar: MPO is increased compared to mock infected constructs and to FSSE peptide pre-treated constructs (top panel). HMGB1 is retained in cells and reduced in construct supernatants of FSSE pretreated constructs compared to those infected but untreated with FSSE (panels 2 and 4). Actin for loading control is shown in panel 3.

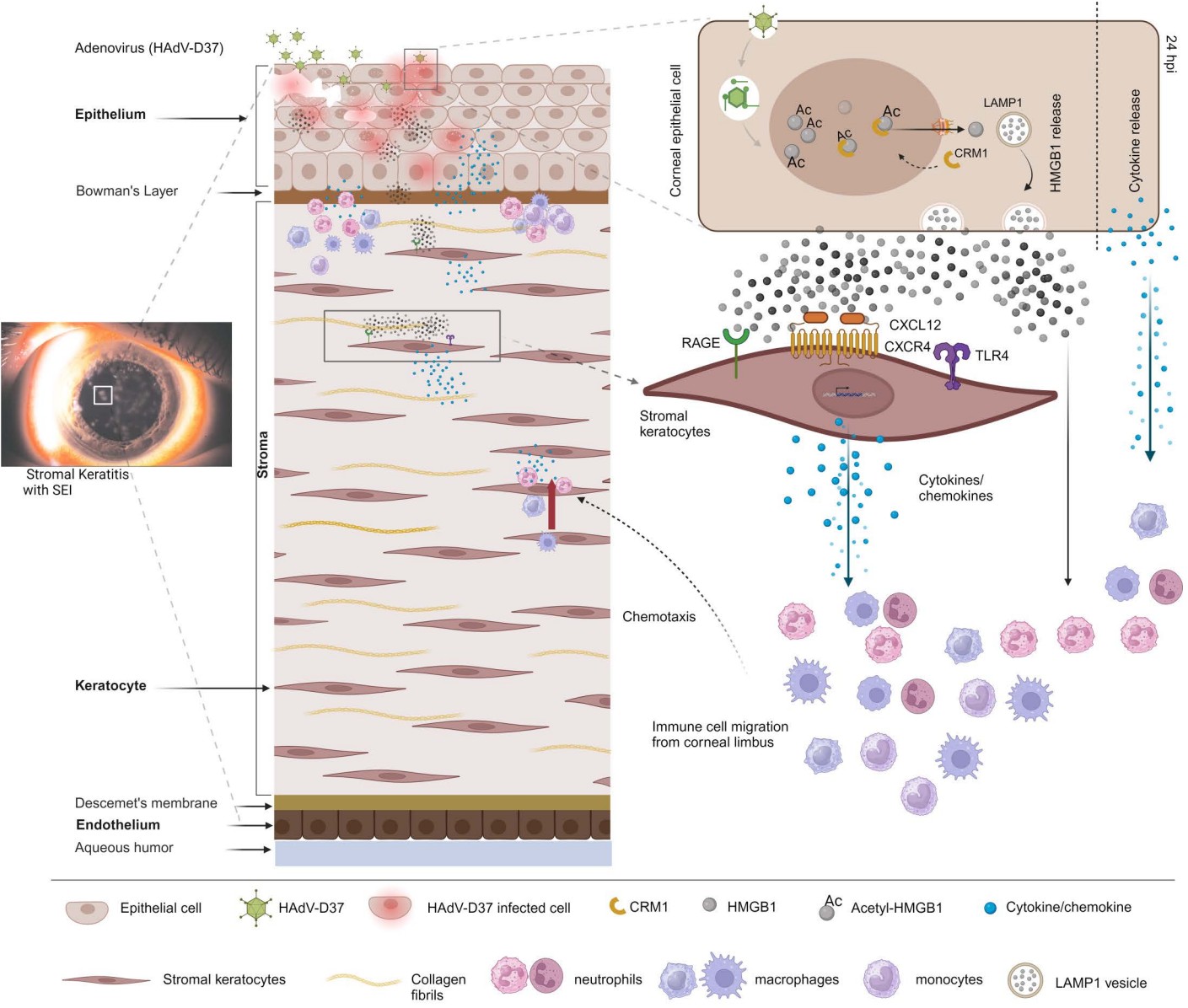

**Fig 9. SEI formation schematic of the study.** Schematic representation (created using Biorender.com) of the mechanism of stromal keratitis/SEI formation upon adenovirus infection. HMGB1 and other mediators expressed by infected corneal epithelium, and the mediators induced by HMGB1 interaction with underlying stromal keratocytes, cumulatively forms a cytokine storm leading to SEI formation, the hallmark of EKC. The human eye photograph with subepithelial infiltrates is reproduced with permission [5] (http://creativecommons.org/licenses/by/4.0/).

reduced myeloperoxidase (MPO) expression in infected facsimiles when pretreated with FSSE (Fig 8D), MPO expression within the constructs increased after infection, regardless of the presence of HCF in the constructs, and the increase was diminished by treatment with FSSE. Facsimile supernatants showed an increase of secreted HMGB1 upon infection, and this was reduced with FSSE pretreatment. These data correlate well with the imaging performed on the constructs. Although facsimile constructs made without stromal cells were not sufficiently stable to enable histology, Western blot analysis showed a similar pattern of HMGB1 and MPO expression with or without the presence of HCF (Fig 8E), suggesting that HMGB1 expression by infected corneal epithelial cells was sufficient for stromal inflammation, and that HMGB1 was acting as a chemoattractant. We also performed immunofluorescence staining for MPO and F4/80 on the constructs with THE cells and HCF, and the immune cell infiltrates were positive for both markers (S7A and S7B Fig). To dissect whether HMGB1 secreted by the infected epithelial cells would have a stand-alone effect on immune cell migration, we treated the constructs with rHMGB1, and tested for the expression of MPO. Compared to mock controls, either viral infection or rHMGB1 treatment alone led to MPO expression. (S7C Fig). A model of HMGB1 expression by corneal epithelial cells is represented in Fig 9.

## Discussion

The epithelial cell barrier at mucosal surfaces serves as a primary line of defense against invading pathogens. Beyond its physical barrier function, mucosal epithelial cell expression of innate immune mediators are by definition among the very first cellular responses to infection. In respiratory epithelium infected by SARS-CoV-2, viral infection initiates expression of type I and III interferons, leading to a cascade of hyperinflammatory responses that ultimately create tissue destruction and loss of respiratory function [81]. In comparison to studies in other tissues, those detailing epithelial cell responses to infection and their mechanisms at the ocular surface have been relatively few [82–85]. Most adenovirus infections are self-limiting, but in EKC corneal inflammation (manifest as SEI) can be prolonged or relapse repeatedly for months to years after infection in up to one-third of patients [8]. Although there have been multiple studies detailing a potential role for keratocytes in the corneal stroma in the induction of SEI after adenovirus infection [14], there is to date no evidence that adenoviruses can invade through the dense stromal extracellular matrix of the cornea to directly infect stromal cells. Therefore, the molecular mechanism(s) behind SEI formation has remained uncertain. Herein, we show expression of the alarmin HMGB1 by infected human corneal epithelial cells, and in a unique 3-dimensional corneal facsimile model that epithelial cell expression of HMGB1 is sufficient for SEI formation.

In airway epithelial cell monolayers infected with HAdV-C5, adenoviral protein VII binds to the A-box of HMGB1 and sequesters HMGB1 in the cell nucleus [26], effectively repressing its alarmin functions [25]. Posttranslational modifications of protein VII were necessary for its binding to HMGB1. However, comparison of the amino acid sequences of protein VII from HAdV-C and HAdV-D showed only ~73% identity, suggesting species-specific function. Importantly, others have demonstrated that acetylation of HMGB1 is the key driver of nuclear to cytoplasm translocation [40,53,86,87]. We show herein that HAdV-D37 infection of both immortalized and primary corneal epithelial cells is associated with HMGB1 acetylation, CRM1-dependent nuclear-to-cytoplasmic translocation, and LAMP1-mediated release of into cell supernatants, in contrast to infection of the same cells by HADV-C5. Nuclear extracts prepared from HAdV-D37 and HAdV-C5 infected THE cells show negligible to no expression of protein VII in the nucleus of the HAdV-D37 infected cells in contrast to the HAdV-C5 infected cells, in which there was a time dependent increase in protein VII expression. These latter data suggest protein VII of HAdV-D37 may function differently during infection to that of HAdV-C5. HAdV-D appears to be a unique HAdV species from the perspective of its evolution [88], and the organization and function of many of its proteins [89]. If protein VII does not accumulate in the nuclei of HAdV-D infected corneal cells, it cannot effectively sequester HMGB1 to prevent its egress.

Using a 3-dimensional corneal facsimile to model natural infection of corneal tissue, infection of corneal epithelial cells by HAdV-D37 was sufficient to induce SEI and the expression of MPO by infiltrating cells, consistent with chemokine

activity. SEI formation and MPO expression in infected corneal facsimiles were both blocked by an HMGB1 inhibitor. Treatment of the uninfected corneal facsimiles with rHMGB1 was sufficient to induce SEI and expression of MPO, showing specificity for HMGB1 in SEI formation. These data strongly implicate HMGB1 as a key mediator of SEI formation after infection of corneal epithelium by cornea-tropic adenoviruses. The data also demonstrates how seemingly canonical molecular pathways, determined in specific cell lines in monolayer culture, do not necessarily translate well to other cell types or to human tissues with different extracellular matrices and varied cellular architectures [39]. For example, we previously demonstrated that the entry and trafficking pathways for HAdV-D37 in human corneal epithelial cells and fibroblasts involved unexpected and non-canonical mechanisms [90,91]. Our results using corneal epithelial cell-tropic HAdVs to infect primary corneal cells strongly suggest that HMGB1 expression is both cell and virus-specific, and further confirm the need to carefully parse disease mechanisms in relevant cell types and experimental models.

HMGB1 release in SARs-CoV-2 infection was recently reported to be both "active", i.e., due to post-translational modifications (acetylation), and "passive", i.e., due to cytolysis at the time of cell death [92]. In reality, all cell death that is not instantaneous, as for example due to direct and sudden trauma to the cell, involves cell signaling and can be considered an "active" process. In our experiments, adenovirus infection of corneal epithelial cells induced HMGB1secretion when LDH release was still below cytopathic levels, and when cells were visually intact upon inspection by microscopy. Infected corneal epithelial cell supernatants also tested negative for pyroptotic pathway components for up to 48 hpi. Although cytolysis of corneal epithelial cells occurs at late stages of adenovirus infection [93], and this may contribute to total extracellular HMGB1 expression, it is acetylation of HMGB1 that is necessary and responsible for nuclear-to-cytoplasmic translocation and subsequent secretion [55,86,87], and it is acetylated HMGB1 that is proinflammatory. Additionally, since adenovirus infection did not alter levels of total HMGB1 mRNA or protein in corneal epithelial cells, it appears to be specifically the post-translational acetylation of HMGB1 that mediates its secretion. CRM1 has been shown to tightly regulate the nuclear-to-cytoplasmic translocation of HMGB1 and is proposed to be the principal pathway for biologically active HMGB1 secretion [31,94]. HMGB1 was previously reported to colocalize with LAMP1 in secretory vesicles in the cytoplasm [56]. Other studies showed that HMGB1 can be secreted by a nonclassical pathway through partially distinct vesicles [30], and that its secretion can be PKR activation dependent [95].

We found that silencing CRM1 expression reduced HMGB1 nuclear-to-cytoplasmic translocation, causing retention of HMGB1 in cell nuclei. Knockdown of LAMP1 led to retention of HMGB1 in the cytoplasm even after 24 hpi, when acetylated HMGB1 would otherwise have appeared in cell supernatants. These particular outcomes are consistent with existing canonical pathways of HMGB1 trafficking [30,31].

Corneal epithelial cell expression of acetylated HMGB1 upon adenovirus infection occurs in a context, and any approach that ignores the expression of other mediators by the same epithelial cells or the impact of HMGB1 on other corneal cells understates the complexities inherent to biologic processes. We also explored the effects of recombinant HMGB1 on corneal stromal cells (HCFs), finding a robust upregulation of proinflammatory cytokines including IL-6, IL-8, MCP-1, MCP-3, and MIF. Upregulation of CXCL12 was particularly notable, because CXCL12 forms a heterocomplex together with HMGB1 and the CXCR4 receptor, creating an axis for signaling and other cellular functions [23]. Our bioinformatics data confirmed a general upregulation of proinflammatory mediators in rHMGB1 treated HCFs, particularly for chemoattractants and proinflammatory cytokines. This is consistent with a role for keratocytes in amplifying a direct chemokine effect of HMGB1 in the cornea. Gene ontology and KEGG analyses further illustrate the potential molecular and cellular pathways induced by the interaction of corneal epithelial cell-derived HMGB1 with underlying stromal keratocytes. RAGE and TLR4 are the best established HMGB1 receptors [96]. When treated with rHMGB1, HCF expression of both RAGE and TLR4 increased as compared to the buffer treated cells, and RAGE expression was increased even relative to treatment with LPS. Further downstream signaling events were induced in HCF by rHMGB1, including phosphorylation of ERK and NFkB, both previously shown to be downstream of RAGE and TLR4 binding [97,98], and reportedly a feature of other quite disparate disease states [99–101]. Further studies will be needed to determine to what degree

HMGB1 expressed by infected corneal epithelial cells penetrates the corneal extracellular matrix to interact with HCF. In infected 3-dimensional corneal facsimiles, HCF were dispensable for SEI formation and MPO expression. Therefore, the impact of HMGB1 on HCF in adenovirus infection is likely additive.

Human tissue infections induce complex and varied cellular responses dependent on the pathogen, the cell type(s) infected, the local tissue architecture, the interactions and molecular cross-talk between the various cell types within the tissue, and the impact of infiltrating immune cells and their byproducts. HAdVs infect in species-specific fashion, and existing *in vivo* models of adenovirus keratitis rely on infection with very high titers of virus and are bereft of viral replication [102,103]. Our results in a 3-dimensional human corneal facsimile model of adenovirus keratitis are consistent with a principal role for corneal epithelial cell-derived, acetylated HMGB1 in the formation of SEI. Stromal infiltration by exogenous immune cells, migrating against gravity, occurred regardless of whether HCF were present in the construct, and also occurred in uninfected constructs exposed to rHMGB1 alone, consistent with a principal role for HMGB1 in adenovirus keratitis. Further study is necessary to determine the specific redox state of corneal epithelial cell-expressed HMGB1. Because infection induced the infiltration of immune cells even in the absence of HCF in the construct, it appeared that the expressed HMGB1 was acting as a chemokine, consistent with a reduced form of the molecule. Alternately, subsequent chemotaxis may have been due to other mediators expressed by infected corneal epithelial cells. However, because rHMGB1 alone induced a similar phenotype of inflammation, it appears that stromal cells are not requisite to the formation of SEI. Such conclusions were possible only with use of the 3-dimensional model, and would have been missed in a monolayer culture system.

In summary, our data suggest that HMGB1 expression by infected corneal epithelial cells is sufficient for corneal stromal inflammation and SEI formation in adenovirus keratoconjunctivitis, representing the missing link between infection of the overlying epithelial cells and subsequent infiltration of the underlying corneal stromal extracellular matrix by immune cells. Suppression of HMGB1 for treatment of sepsis, ischemia, cancer, and autoimmune disorders is currently the focus of multiple clinical trials applying a range of unique HMGB1 antagonists [104], suggesting significant potential for translation to patients with adenoviral keratoconjunctivitis.

## Method details

### Cell culture and virus preparation

Primary HCF were isolated and pooled from donor corneal tissue as previously described [83]. Corneal epithelium and endothelium were removed by mechanical debridement, then the corneas were cut into 2 mm-diameter segments and placed in individual wells of six-well tissue culture plates (Corning, 3516) with DMEM supplemented with 10% FBS, penicillin G sodium, and streptomycin sulfate at 37°C in 5% $CO_2$. After 6 weeks, confluent HCFs are trypsin treated and transferred to T75 flasks for expansion to be used at 3rd passage. PCEC were purchased from MilliporeSigma (SCCE016), and THE cells were the kind gift of Jerry Shay (University of Texas Southwestern Medical Center). PCEC and THE cells were grown in Defined Keratinocyte-SFM (1X) basal media with 0.2% growth supplement (Thermo Fisher Scientific, 10744019) and Keratinocyte Serum-Free Growth Medium for adult cells (Sigma, 131-500A) respectively with 1% Penicillin-Streptomycin solution (Thermo Fisher Scientific, 15-140-122). Primary PCECs were used at 2nd passage. A549 (CCL-185), a human lung carcinoma cell line, and HEK293 (CRL-1573), a human embryonic kidney cell line, were purchased from American Type Culture Collection (ATCC). A549 and HEK293 cells were maintained in DMEM (high glucose) supplemented with 10% FBS (Gibco, Thermo Fisher Scientific, 10082147), 1% penicillin G sodium, and 100µg/ml streptomycin sulfate at 37°C in 5% $CO_2$. Human PBMC were purchased from ATCC and were thawed and cultured in RPMI 1640 Medium (Gibco, 11875093), 12 h prior to use. HAdV types C5, D9, D37, and D56, were a obtained from ATCC and grown in A549 cells in DMEM with 2% fetal bovine serum, and 1% penicillin streptomycin solution. Virus was purified from A549 cells after 7 days of infection using CsCl gradient ultracentrifugation, and the purified virus was titered in triplicate, and stored at -80°C. Purified virus was tested for endotoxin (GenScript, L00350) prior to use.

PLOS Pathogens

## Western blotting and densitometry

Cytoplasmic and nuclear extracts were prepared from cells that were mock treated or infected with HAdV-D37 using NE-PER Extraction Reagent (Thermo Fisher Scientific, 78833). Protein concentrations were determined by BCA Protein Assay (Bio-Rad, 500-0202). Equal amounts of protein were run on a gradient gel (Thermo Fischer Scientific, 4–20% Tris-acetate protein gel, 4561094), and transferred to nitrocellulose membranes (Bio-Rad, 1620115). Membranes were blocked with 5% BSA (Sigma, A9647-500G), and then immunoblotted for HMGB1 (Abcam, ab18256), with both cytoplasmic and nuclear loading controls, β-actin (Abcam., ab8227) and TBP (Abcam, ab74222), respectively. To determine infectivity, both adenoviral late protein pIIIa antibody (kind gift from Dr. Patrick Hearing, Stony Brook University) and adenovirus type 5 pan-antibody (Abcam, ab6982) were used. Bands visualized by chemiluminescence (Thermo Fisher Scientific, Supersignal West Dura, 34075) were analyzed by densitometry using ChemiDoc (Bio-Rad).

## Real-time PCR

Total RNA was extracted from mock and HAdV-D37 infected cells with a RNeasy Minikit (Qiagen. 74104). RNA samples were quantified using a NanoDrop spectrophotometer (Thermo Scientific). Synthesis of cDNA was performed with 1 μg of total RNA in a 20-μl reaction mix using oligo(dT) and Moloney murine leukemia virus (M-MLV) reverse transcriptase (Promega, M1701a). Quantitative real-time PCR (qRT-PCR) amplifications (performed in triplicate) were done with 1 μl of cDNA in a total volume of 20 μl using Applied Biosystems Fast SYBR Green master mix (Thermo Fisher Scientific, 4385612). The forward and reverse primers for HMGB1 were 5'-GCGAAGAAACTGGGAGAGATGTG-3' and 5'-GCATCAGGCTTTCCTTTAGCTCG-3', respectively. Viral DNA replication was determined by expression of E1A. The forward and reverse primers for E1A were 5'-CGCCTCCTGTCTTCAACTG-3' and 5'-TGGGCATCTACCTCCAAATC-3' respectively. 18S RNA was used as the housekeeping gene control for normalization. PCR assays were run using the QuantStudio 3 system (Applied Biosystems) under the following conditions: 95°C (10 seconds), 60°C (1 min), and 72°C (30 seconds) for 40 cycles, followed with a final extension at 72°C for 10 min. Data was analyzed by the comparative threshold cycle (CT) method. Each experimental condition was analyzed in triplicate wells and repeated three times.

## Confocal microscopy

Cells grown on chamber slides (Thermo Fisher Scientific, 177437) were infected at an MOI of 1. Cells were then fixed in 4% paraformaldehyde for 10 min, washed in PBS containing 1% BSA, and permeabilized for 10 min in solution containing 0.1% Triton X-100, followed by three washes in 1x PBS containing 1% FBS. After 30 min blocking in 2% FBS-PBS, cells were incubated in antibody to acetyl-HMGB1 (Aviva Systems Biology, OASG03545) overnight at 4°C, followed by three washes in 1x PBS containing 1% FBS. Cells were then further incubated with Alexa Fluor 488 conjugated secondary antibody (1:1000, Thermo Fisher Scientific) for 45 min at room temperature. For actin staining, the cells were incubated in 1:1000 of Alexa Fluor 568 phalloidin (Thermo Fisher Scientific) for 30 min at room temperature, and washed three times in 1x PBS containing 1% FBS. Washed cells were mounted using Vectashield mounting medium (Vector Laboratories, H-1200-10) containing DAPI. Images were captured with a Leica SP5 con-focal microscope using a 63x oil immersion objective. Images were scanned at 0.5 μm intervals to obtain 15–20 Z-stacks each, and an image from the middle stack represented.

For analysis of colocalization for HMGB1 with CRM1 and LAMP1, THE cells infected with HAdV-D37 for 12 h were processed for confocal microscopy as above. Cells were incubated with HMGB1 antibody (Abcam, ab18256, 1:1000) for 12 h followed by anti-rabbit Alexa Fluor 488 secondary antibody (1:1000) for 1 h at room temperature. Cells were then incubated with CRM1 (Mouse CRM1, Santa Cruz Biotechnology, sc74454, 1:500) or LAMP1 (Mouse LAMP1, Santa Cruz Biotechnology, sc20011, 1:500) antibody for 2 h at room temperature followed by incubation with appropriate secondary antibody (anti-mouse Alexa Fluor™ 568, Thermo Fisher Scientific A11004, 1:500) for 1 h at room temperature.

## High content microscopy

Cells were cultured in 12 well plates (Costar, 5313) and infected for 6, 8, 10, 12, and 24 h at MOI 5 (with this MOI chosen for these experiments to best enable visualization of HMGB1 secretion over time), fixed with 4% paraformaldehyde for 10 min, washed in PBS containing 1% fetal bovine serum (FBS) and permeabilized with 0.1% Triton X-100. Infected cells were then blocked in 2% FBS-PBS for 30 min followed by incubation with HMGB1 antibody (Abcam, ab18256, 1:1000) for 12 h. Secondary antibody (anti-rabbit Alexa Fluor 488, Abcam, ab150077, 1:1000) incubation was performed for 2 h at 37°C and washed in PBS containing 1% FBS 3 times. For actin staining, cells were incubated in 1:1000 of Alexa Fluor 568 phalloidin (Thermo Fisher Scientific, A12380, 1:1000) for 30 min at room temperature and washed three times in 1x PBS containing 1% FBS followed by DAPI staining (Sigma, D9542-10MG). High content microscopy with automated image acquisition and quantification was carried out using a Cellomics HCS scanner and iDEV software (Thermo Fisher Scientific). For HCM analyses, > 10,000 primary objects were counted per well, and a minimum of 3 wells per condition were counted in each experiment. The data presented are derived from 3 or more independent experiments.

## siRNA knockdown

Silencer Select Non-targeting negative control siRNA (4390843), CRM1 siRNA (14937), and LAMP1 siRNA (4392420, ID: s8080) were all obtained from Ambion. Briefly, cells were seeded at 24 h before transfection, and 50 pmol of each siRNA was transfected using Lipofectamine RNAiMAX (Invitrogen, 13778150) in Opti-MEM reduced serum medium (Gibco, 31985070). After 24 h, cells were treated again with siRNAs for a second round of knockdown. At 48 h post transfection, cells were infected with purified virus at a MOI of 5 for 2, 12, 24, and 48 h. Cell-free supernatants were collected for Western blot. Cells at 12 and 24 h were processed for HCM as above with antibodies to LAMP1 (Abcam, ab278043, 1:1000) and CRM1 (Abcam, ab24189, 1:1000).

## Cell viability and LDH assay

THE or PCEC cells were seeded in 96 well plates (10,000 cells per well). The following day, the cells were infected at an MOI of 1. Supernatants were assayed for LDH using an LDH cytotoxicity colorimetric assay kit (Thermo Scientific, Pierce, PI88953) as per the manufacturer's instructions, and repeated at least three times. To rule out cellular toxicity of FSSE, different concentrations of FSSE were added in 8 replicates per concentration to THE or PCEC cells in 96-well plates at 80–90% confluency. Cytotoxicity was analyzed by MTT assay (Abcam, ab211091).

## Pyroptosis and secretion assay

THE and PCEC were seeded in 6-well plates and mock infected or infected with HAdV-D37 for 2, 12, 24, and 48 h at MOI 1. At the end of each incubation, cell culture supernatants were collected and cleared of debris by centrifugation at $500 \times g$ for 5 min. Protein concentrations were measured and equal amounts of proteins resolved, transferred to nitrocellulose membrane, and immunoblotted with the antibodies to IL-18 (R&D Systems, D043-3, 1:1000), IL-1β (R&D Systems, MAB201, 1:1000), GAPDH (Abcam, ab181602), and HMGB1 (Abcam, ab18256, 1:1000).

To confirm antibody reactivity, THE cells were treated with or without LPS (Cell Signaling, 14011) in regular culture media (2μg/ml for 8h). After treatment, cell culture supernatants were collected and cleared of debris, washed once with PBS, and lysed in lysis buffer with protease inhibitor (Cell Signaling, 5871). Western blot analysis was performed as described above with antibodies to IL-18 (R&D Systems, D043-3, 1:1000) and IL-1β (R&D Systems, MAB201, 1:1000). β-actin (Abcam, ab8227) expression was also measured as control for equal loading.

## Cytokine array

Cultured HCF at near confluence were placed in 2% FBS overnight, and then treated with 2μg/ml recombinant human HMGB1 (R&D Systems, 1690-HMB-050) or PBS control for 4 hours. In separate experiments, THE and PCEC were

infected with HAdV-D37 at an MOI of 1 for 24 hr. Cell free supernatants from each cell type and experimental condition were collected and processed with the Human XL Cytokine Array Kit (R&D Systems, ARY022B) with antibodies to 105 different proteins in duplicate on the array membranes, as per the manufacturer's instructions. Briefly, array membranes were incubated for 60 min in 2 ml of blocking buffer on a shaker at room temperature. 1 ml of supernatant from each group/condition was incubated with 500 µl reconstituted human cytokine array detection cocktail for 60 min, and then placed on the array membranes overnight at 4° C. Following a washing step, the membranes were incubated with a 1:2000 dilution of streptavidin-conjugated peroxidase for 45 min at room temperature. Proteins were detected by SuperSignal™ West Dura Extended Duration Substrate enhanced chemiluminescence (Thermo Scientific, 34075), and signals were captured on a BioRad ChemiDoc Image Station and analyzed in ImageJ. The normalized densitometry values obtained from dot blots were converted to z-scores. This was done by subtracting each data point from the mean of its respective row (which is the average of all data points in a row) and then dividing the result by the standard deviation of the row. A heat map was generated using Prism (GraphPad Software, v.8.0.1), where the z-score values were used to determine the color range. Each experiment was repeated at least three times.

To model the possible impact of HMGB1 on underlying corneal stromal cells, HCF were treated with rHMGB1 (R&D Systems, 1690-HMB-050, 2µg/ml), LPS (Cell Signaling, 14011, 2µg/ml), or mock treated with PBS. Total cell lysates were prepared after 4 h for Western blot to assess TLR4 (Abcam, ab13556, 1:1000), RAGE (Abcam, ab216329, 1:1000), CXCR4 (Abcam, ab124824, 1:1000), NF-kB (Cell Signaling Technology, 8242, 1:1000)/p-NF-kB (Cell Signaling Technology, 3033, 1:1000), ERK (Abcam, ab184699, 1:1000)/p-ERK (Abcam, ab201015, 1:1000).

### Sequence analysis

Reference sequences for HAdV-C and D were aligned using CLUSTALW, and the protein VII amino acid differences were analyzed in BioEdit Sequence Alignment Editor (v7.2.5).

### STRING analysis

STRING (version 12.0; http://string-db.org) was used to analyze the functional interactions between cytokines and visualize protein-protein interaction (PPI) networks. The threshold of protein interaction was set to medium confidence (default setting). Enrichment analysis of identified cytokines, including Gene Ontology (GO) functional analysis (biological process, molecular function, and cellular component) and Kyoto Encyclopedia of Genes and Genomes (KEGG) pathway analysis, was performed using ShinyGO 0.80 (http://bioinformatics.sdstate.edu/go/). Graphs were prepared using Prism.

### 3D corneal construct fabrication and immune cell migration assay

3D corneal constructs were generated as recently reported [80] in 12mm diameter cell culture membrane inserts (Coster, 3402) with 3µm pores. Collagen type I (3–4 mg/ml, Corning, 354236), collagen buffer (10X HEPES:10X DMEM: FBS = 1:1:1.11), and chondroitin sulfate (12:1 v/v collagen) were combined and neutralized with 1N NaOH, and then crosslinked with aqueous glutaraldehyde (0.022%) with unreacted aldehyde groups further neutralized by glycine solution (1.66%). $3x10^6$ HCF were mixed into 1.5 ml of the crosslinked mixture, and 600 µl was poured into the insert followed by the incubation at 37°C for get formation. $6.5x10^5$ THE cells were seeded onto the collagen gel surface and the insert was maintained with DMEM/Ham's F-12 (Corning, 10-092-CV) with 10% fetal calf serum and 10 ng/ml EGF for 5 days. The constructs were treated with rHMGB1 or infected at MOI 15 for 12 h with or without FSSE (Biomatik Peptide Synthesis: P5779), and PBMCs (ATCC, PCS-800-011) were added to the bottom of each insert and incubated for 4 h. Viability of the PBMCs was assessed by trypan blue staining prior to each experiment, and in each instance found to be > 98%. The constructs were either fixed in 4% paraformaldehyde and processed for paraffin embedding, or lysed in lysis buffer (Cell Signaling, 9803) with protease inhibitor cocktail (Cell Signaling, 5871) followed by ultrasonic homogenization (Bransonic, 1510R-DTH) for Western blot analysis.

## Immunohistochemistry

Tissue sections were deparaffinized using xylene, and the samples rehydrated in water through a graded series of alcohols (100%, 96%, 70%, 50%, and water). For antigen retrieval, sections were incubated overnight in 10 mM sodium citrate buffer, 0.05% Tween 20 (pH 6.0) at 60°C, washed with Tris-buffered saline (TBS) plus 0.025% Triton X-100, followed by blocking unspecific binding using TBS supplemented with 10% FBS and 1% bovine serum albumin (BSA). The sections were then incubated with primary antibodies against MPO (Abcam, ab208670; 1:100) or F4/80 (Thermo Fisher Scientific, MA1–91124, 1:100) overnight at 4 °C. Incubation with secondary antibodies anti-rabbit IgG Alexa Fluor 647 (1:100) and anti-rat IgG Alexa Fluor 594 (1:100), for MPO and F4/80, respectively, was carried out for 1 h at room temperature. Slides were mounted in VectaShield mounting medium containing DAPI (Vector Laboratories, H-1200-10) and photographed with a Zeiss LSM510 laser scanning microscope with a 20x objective.

## Statistical analyses

All assays were performed a minimum of three times with each sample in triplicate. Results were graphed showing the means and error bars for standard deviations. Data were analyzed with a paired two-tailed Student's *t*-test or by one-way ANOVA with Tukey multiple comparison test. A p value < 0.05 was considered statistically significant. Analyses were performed using GraphPad Prism v6.0 (GraphPad Software). All values were normalized using the maximum-minimum method and a 0–1 scale. Asterisks indicate statistical significance (* < 0.05, ** < 0.01, *** < 0.001). The minimal data set for all the graphs is attached as S1 Table.

## Supporting information

**S1 Fig.  Host and viral gene and protein expression in corneal epithelial cells infected with HAdV-D37.** (A) Western blot analysis for HMGB1 expression along with β-actin for load control from whole cell lysates of uninfected (M) or HAdV-D37-infected (V) THE cells for 2, 12, 24, and 48 hpi. qRT-PCR analysis of HMGB1 gene expression for mock and virus infected cells at the same times pi is shown as bar graphs below the Western blot. (B) Bar graph for qRT-PCR of the viral early gene E1A expression, a surrogate marker for viral entry, and normalized to human ACTG gene for quantification. (C) Viral late protein pIIIa expression in cytoplasmic and nuclear extracts prepared from uninfected or HAdV-D37-infected THE, PHCE, HCF, A549 and HEK293 for 2, 12, 24, and 48 hpi show successful infection of all cell types.
(TIF)

**S2 Fig.  Time dependent release of HMGB1 in HAdV-37 infection.** (A) Cytoplasmic HMGB1 localization compared between mock and viral infection at indicated times pi. (B) HMGB1 distribution between the nucleus and cytoplasm compared between nucleus and cytoplasm within each group at various time point of infection.
(TIF)

**S3 Fig.  Omics analysis of immortalized human corneal epithelial cells infected with HAdV-D37.** (A) Human cytokine array performed on cell supernatant from THE mock treated with dialysis buffer (upper panel) or HAdV-D37 infected (lower panel) at 24 h. (B) Upregulated proteins compared graphically using ImageJ quantification, and compared to same protein in mock infection. (C) The physical and functional associations among the upregulated proinflammatory mediators were assessed using the STRING tool. The interaction among the query proteins represents the network with 10 nodes and 21 edges of protein-protein interaction (PPI). (D-G) Gene ontology (GO) showing top biological process, molecular function, KEGG pathway, and cellular components. Bar length represent the number of genes and dotted line represents -log10 adjusted p value for significantly enriched pathways.
(TIF)

**S4 Fig. Omics analysis of primary human corneal epithelial cells infected with HAdV-D37.** (A) Human cytokine array performed on cell supernatants of, PCEC mock infected (upper panel) or infected with HAdV-D37 (lower panel) at 24 hpi. (B) Upregulated proteins compared graphically using ImageJ quantification, and compared to the corresponding protein in mock infection. (C) The physical and functional associations among the upregulated proinflammatory mediators were assessed using the STRING tool. The interaction among the query proteins represents the network with 11 nodes and 11 edges of protein-protein interaction (PPI). (D-G) Gene ontology (GO) showing top biological process, molecular function, KEGG pathway, and cellular components. Bar lengths represent the number of genes and dotted lines represents -log10 adjusted p values for significantly enriched pathways. (H) Venn diagram showing the upregulated, overlapping cytokine expression between HCF, THE, and PCEC.
(TIF)

**S5 Fig. Role of CRM1 and LAMP1 in HMGB1 trafficking.** (A) THE cells treated with NC-siRNA or CRM1-siRNA, and HAdV-D37 infected at a MOI of 5 for 2, 12, 24, and 48 h. Cell-free supernatants and total cell lysates were prepared for Western blot to measure HMGB1 and CRM1 knockdown, respectively. β-actin was used as an internal loading control ($n = 3$). (B) Immunofluorescence for DAPI nuclear (blue), CRM1 (red), and HMGB1 (green) in mock treated and HAdV-D37 infected cells at an MOI of 5 at 12 hpi. Colocalization between HMGB1 and CRM1 (yellow nucleus) was seen only in virus infected cells. Scale bar = 10 μm. (C) THE cells were treated with NC-siRNA and LAMP1-siRNA, and HAdV-D37 infected at an MOI of 5 for 12 and 24 h. Cell-free supernatants and total cell lysates were prepared to measure HMGB1 and LAMP1 knockdown respectively by Western blot. β-actin used as an internal loading control ($n = 3$). (D) Immunofluorescence showing nuclei (blue), LAMP1 (red), and HMGB1 (green) in mock infected and HAdV-D37 infected cells (MOI of 1 and 10, at 12 hpi). Scale bar = 10 μm. ($n = 5$). (E) HCM analysis of THE cells infected with HAdV-D37, and immunostained for HMGB1 and LAMP1 and analyzed for colocalization. Data shown as the mean ± SD ($n = 3$). ANOVA with Tukey's post-hoc test was performed. For HCM, > 5000 cells were counted per well, with a minimum number of 3 valid wells ($n = 3$).
(TIF)

**S6 Fig. 3D culture mimics natural disease at 24hrs post-infection.** Photomicrographs depicting 3D corneal constructs at 24 hpi. H&E staining of the mock infected corneal construct with PCEC and THE shows a healthy stratified epithelium and stroma with no PBMC migration (column 1, panels 1 and 2, respectively). HAdV-D37-infected corneal constructs show foci of immune cell migration as indicated by white arrows (column 2, panels 1 and 2). DAPI staining is shown for constructs from the same groups, and demonstrates immune cell foci in the setting of HAdV-D37 infection (column 2, panels 3 and 4).
(TIF)

**S7 Fig. MPO and F4/80 expression in infected 3D corneal constructs.** (A) Sections from 3D corneal constructs were stained with DAPI, and immunostained for MPO expression. (B) Sections stained with DAPI and immunostained for F4/80 expression. Figures are representative of images from three 3D corneal constructs. (C) Western blot analysis showing expression of MPO in constructs infected with HAdV-D37 or treated with rHMGB1.
(TIF)

**S1 Table. Minimal data set.**
(DOCX)

# Acknowledgments

We extend our sincere thanks to Sharina P. Desai, UNM AIM Core Technical director for helping with data acquisition.

## Author contributions

**Conceptualization:** Amrita Saha, Jaya Rajaiya.

**Data curation:** Amrita Saha, Mohammad Mirazul Islam, Rahul Kumar, Ashrafali Mohamed Ismail, Jaya Rajaiya.

**Formal analysis:** Amrita Saha, Jaya Rajaiya.

**Funding acquisition:** Mohammad Mirazul Islam, James Chodosh, Jaya Rajaiya.

**Investigation:** James Chodosh, Jaya Rajaiya.

**Methodology:** Amrita Saha, Mohammad Mirazul Islam, Rahul Kumar, Ashrafali Mohamed Ismail, Emanuel Garcia, Jaya Rajaiya.

**Project administration:** Jaya Rajaiya.

**Resources:** Rama R. Gullapali, Jaya Rajaiya.

**Supervision:** James Chodosh, Jaya Rajaiya.

**Validation:** Jaya Rajaiya.

**Visualization:** Jaya Rajaiya.

**Writing – original draft:** Jaya Rajaiya.

**Writing – review & editing:** Amrita Saha, James Chodosh, Jaya Rajaiya.

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
