## [Decision Letter · Decision Letter 0]

5 Mar 2025

PPATHOGENS-D-25-00122

Virus and Cell Specific HMGB1 Secretion and Subepithelial Infiltrate Formation in Adenovirus Keratitis

PLOS Pathogens

Dear Dr. Rajaiya,

Thank you for submitting your manuscript to PLOS Pathogens. After careful consideration, we feel that it has merit but does not fully meet PLOS Pathogens's publication criteria as it currently stands. Therefore, we invite you to submit a revised version of the manuscript that addresses the points raised during the review process.

Please submit your revised manuscript within 60 days. If you will need more time than this to complete your revisions, please reply to this message or contact the journal office at plospathogens@plos.org. Please include the following items when submitting your revised manuscript:

We look forward to receiving your revised manuscript.

Kind regards,

Donna M Neumann

Academic Editor

PLOS Pathogens

Robert Kalejta

Section Editor

PLOS Pathogens

 Sumita Bhaduri-McIntosh

Editor-in-Chief

PLOS Pathogens

orcid.org/0000-0003-2946-9497

 Michael Malim

Editor-in-Chief

PLOS Pathogens

orcid.org/0000-0002-7699-2064

**Additional Editor Comments:**

The reviewers requested additional chromatin fractionation experiments to show the localization of HMGB1 (cytoplasm, nucleus) and a timecourse for the 3D infections. In addition, please be sure to provide the requested loading controls for experiments in the manuscript as well as a better and more representative image for Figure 3F. Please be sure to address all of the comments made by all three reviewers in the revised manuscript.

**Journal Requirements:**

https://journals.plos.org/plospathogens/s/submission-guidelines#loc-parts-of-a-submission

- ® on page: 35 and 36.

5) We notice that your supplementary Figures are included in the manuscript file. Please remove them and upload them with the file type 'Supporting Information'. Please ensure that each Supporting Information file has a legend listed in the manuscript after the references list.

Potential Copyright Issues:

- Please confirm that you are the photographer of Figures 8A and 9., or provide written permission from the photographer to publish the photos under our CC BY 4.0 license.

- Figure 9. Please confirm whether you drew the images / clip-art within the figure panels by hand. If you did not draw the images, please provide a link to the source of the images or icons and their license / terms of use; or written permission from the copyright holder to publish the images or icons under our CC BY 4.0 license. Alternatively, you may replace the images with open source alternatives. See these open source resources you may use to replace images / clip-art:

7) We note that your Data Availability Statement is currently as follows: "All data are available without restriction". Please confirm at this time whether or not your submission contains all raw data required to replicate the results of your study. Authors must share the “minimal data set” for their submission. PLOS defines the minimal data set to consist of the data required to replicate all study findings reported in the article, as well as related metadata and methods (https://journals.plos.org/plosone/s/data-availability#loc-minimal-data-set-definition).

- The points extracted from images for analysis..

8) Please ensure that the funders and grant numbers match between the Financial Disclosure field and the Funding Information tab in your submission form. Note that the funders must be provided in the same order in both places as well.

**Reviewers' Comments:**

Reviewer's Responses to Questions

**Part I - Summary**

Reviewer #1: Saha et. al., present the results of a study focusing on mechanisms involved in the formation of corneal infiltrates due to Adenovirus group D viruses. These infiltrates interfere with vision resulting in substantial costs. They hypothesize that infection of corneal epithelial cells results in the release of pro-inflammatory mediators that then act on stromal keratocytes to promote inflammation. This could explain the formation of the infiltrates in the absence of stromal cell infection. They focus on HMGB1 based on previous work. They used various continuous cell lines as well as primary corneal epithelial cells and primary stromal keratocytes and a novel 3D corneal cell culture system. The data indicate cell and virus specific effects on HMGB1 expression and release from cells and show that an Adenovirus C (not associated with ocular infection) behaves very differently in terms of affecting HMGB1. Cytokine profiles are compared and showed a number of pro-inflammatory mediators are induced by HGB1 treatment in stromal cells. In the 3D system they demonstrate that HMGB1 treatment results in chemotaxis of neutrophils into the corneal stroma. Overall the work is comprehensive, well designed, and the data support the conclusions. There are a couple of minor issues that should be considered.

1. The opening sentence of the introduction could be deleted since is it not directly relevant. Just start with the Adenovirus introductory material.

2. In Figure 8 panel C the IF photos of the infiltration are not very convincing and not quantitative. They provide quantitation using blotting for meyeloperoxidase, however. It might be more convincing to include additional panels or insets at higher magnification showing the infiltrates into the stroma.

Reviewer #2: The is a generally well-written manuscript describing mechanistic studies on a common, virulent and highly contagious type of corneal infection. Specifically, the authors describe a novel molecular mechanism that leads to corneal subepithelial infiltration (SEI) formation, a key event in this infection. The research findings add valuable new insights into the pathogenesis of adenovirus-induced keratitis, and present a new understanding of the immunological processes involved with potentially broader implications.

The authors provide substantial evidence that secretion of biologically active HMGB1 by infected corneal cells is specific both to the viral agent and the infected cell type. They show that HAdV-D37 infection of immortalized as well as primary corneal epithelial cells are associated with HMGB1 acetylation, CRM1-dependent nuclear-to-cytoplasmic translocation, and LAMP1-mediated release into cell supernatants, in contrast to infection of the same cells by HADV-C5. An important strength of this study is using the 3-dimensional corneal facsimile to simulate and model natural infection of corneal tissue in addition to a monolayer model.

Although broadly well-designed and supported, several aspects of the manuscript could be improved for clarity and accuracy.

Reviewer #3: In this study, Saha et al. have examined how Human Adenovirus D regulates HMGB1 to control epidemic keratoconjunctivitis (EKC). They show that HAdV-D induces cell-specific secretion of HMGB1 which is regulated by CRM1 and LAMP1-mediated cellular signals. They then establish a new 3D culture system that mimics corneal architecture, exhibiting HMGB1 release that induce leukocytic infiltrates. These findings uncover new insights into the biology of HAdV, establish new physiologic platforms to mechanistically investigate HAdV biology and develop inhibitors to HAdV infection. The authors present their findings in a logical and easy-to-follow manner, with schematics summarizing the findings of each figure, making the manuscript easy to follow. While the findings of this study are potentially exciting, they will benefit from some orthogonal validation assays and characterization of their new infection models, which are described in the Comments.

**Part II – Major Issues: Key Experiments Required for Acceptance**

Reviewer #1: None.

Reviewer #2: None noted

Reviewer #3: Fig. 5 – It is unclear why GAPDH is not observed in these samples. Regardless, these western blots require a loading control. The LPS-treated samples should be run with AdV infected THE cells to show the dynamic range of IL1b and IL18 induction. More concerning, however, is that in Fig. 5C the antibody does not seem to detect significant induction of IL1b and IL18 activation by LPS. This discrepancy should be resolved.

Chromatin fractionation studies are needed to determine whether nuclear HMGB1 is sequestered on host chromatin in the presence of protein VII from HAdV-D37 or whether it is remaining in a soluble state in the nucleoplasm. Along those lines, HAdV-D37 protein VII-HMGB1 coIP’s would also be beneficial for demonstrating direct interactions of HAdV-D protein VII with host HMGB1.

The findings of the siCRM1/siLAMP studies should be orthogonally validated by fractionation experiments combined with immunoblots to determine whether the HMGB and viral protein VII is associated with host chromatin, in the nucleoplasm or cytoplasm.

A timecourse of the infection of the 3D culture is needed to provide a proper characterization of this exciting new system of viral infection.

Fig. 3F – the western blot should be rerun to resolve protein VI and VII more clearly.

**Part III – Minor Issues: Editorial and Data Presentation Modifications**

Reviewer #1: See the comments above.

Reviewer #2: General:

The underlying rationale for the choice of MOI used in various experiments, or the potential impact if any of using different levels, is not well explained. For example, in figures 3A and 4B, cell viability and LDH assay, confocal microscopy (line 631) and pyroptosis assay (line 685) used HAdV-D37 infection of cells at an MOI of 1, whereas high content microscopy (line 652) and siRNA lockdown (line 671) used MOI5. Why the choices and do those differences matter?

A strength of the paper is the multiplicity of orthogonal approaches taken to discover and define the unique interactions that occur in their infection model, but a weakness is in the logical flow in which these are presented. In some cases, discussion of the results and comparison to other studies would better be reserved for the Discussion section. Specifically, the presentation from the first results “HMGB1 is released by adenovirus infected cells in cell-specific fashion” to the “HMGB1 acetylation is cell and virus specific” flows fairly smoothly and the logic is clear. However, the transition into “HMGB1 secretion upon viral infection is independent of pyroptosis” would benefit from some explanation as to why this is the next logical question to flow from those findings since other mechanisms may also impact HMGB1 secretion.

Specific:

Lines 114-119: These are not part of the results but could be of greater value in the introduction to explain lines 101 -102 about why potential crosstalk between tissue-specific epithelial cells and fibroblasts is a key issue.

Lines 163-164: Why were THE rather than PCEP cells chosen to elucidate the timeline of HMGB1 secretion upon HAdV-D37 infection here?

Line 205: Fig 3b should be Fig 3B for consistency

Lines 206-207: “HMGB1 cytoplasmic translocation occurred with infection by HAdV-D9 (Fig 3B)”. Figure 3B shows that this is also true for HAdV-D37 and HAdV-D56?

Lines 212-214/Figure 3D: Did the authors use a reference sequence to make the phylogenetic tree or an outgroup?

Lines 218-219/Figure 3E: “The protein VII amino acid identity and similarity scores comparing HAdV-D (types 9, 37, and 56) and HAdV-C (types 2 and 5) ranged between 72.41 to 73.63% and 79.31 to 80.60%, respectively (Fig 3E)” but only the comparison between D37 and C5 is shown in Figure 3E. Potentially, description of the % identity in the text may be enough without the figure.

Lines 274-284: This section could be clearer and more concise. Since Figure 5A shows a schematic representation of corneal epithelial cell release of HMGB1 based largely on what is known from the literature, highlighting many of the mechanisms that influence HMGB1 secretion, this would seem to be an ideal opportunity to explain why the authors then focused on pyroptosis.

Lines 327-328: Are the authors implying a distinction between CRM1 and exportin 1 XPO? If not, it would be better to stick to one nomenclature throughout.

Lines 353/Figure 6: Although the authors mentioned that “HMGB1 was previously shown to partially colocalize in LAMP1 containing vesicles”, there are other pathways and molecules involved not tested that should be considered in the Discussion.

Lines 361-362/Figure 6G: As mentioned above, there are other pathways and molecules involved that are not tested. Figure 6G seems to imply that only LAMP1 is involved, which should be clarified.

Lines 375-377/Figure E-H: Authors perform substantial bioinformatic analysis, however the main outcomes of that analysis could be better explained. Further it detracts that the text in these figures is blurred.

Line 477, references 80-83: Are there more recent studies that bear on this point?

Line 583: Typo, should be C02

Lines 592-593: streptomycin 100g/ml is likely missing a mu.

Figures:

Figure 3G: I did not see a reference to this in the text. The figure may not be essential.

Figure 4C: Scale and control are missing on the WB.

Figure 4E: What new results does this figure provide?

Figure 7C (as well as E, G and H): text is difficult to read.

Supplemental Figures:

S1C: Please add scale and positive control in the WB.

SD-G: Text in these figures seems blurred.

Reviewer #3: The manuscript would benefit from thorough editing for grammatical correctness.

PLOS authors have the option to publish the peer review history of their article (what does this mean? ). If published, this will include your full peer review and any attached files.

**Do you want your identity to be public for this peer review?** For information about this choice, including consent withdrawal, please see our Privacy Policy .

Reviewer #1: No

Reviewer #2: No

Reviewer #3: No

**Figure resubmission:**
---

## [Editor Report · Decision Letter 1]

2 May 2025

Dear Dr. Rajaiya,

We are pleased to inform you that your manuscript 'Virus and Cell Specific HMGB1 Secretion and Subepithelial Infiltrate Formation in Adenovirus Keratitis' has been provisionally accepted for publication in PLOS Pathogens.

Best regards,

Donna M Neumann, PhD

Section Editor

PLOS Pathogens

Robert Kalejta

Section Editor

PLOS Pathogens

Sumita Bhaduri-McIntosh

Editor-in-Chief

PLOS Pathogens

orcid.org/0000-0003-2946-9497

Michael Malim

Editor-in-Chief

PLOS Pathogens

orcid.org/0000-0002-7699-2064
---

## [Editor Report · Acceptance letter]

Dear Dr. Rajaiya,

We are delighted to inform you that your manuscript, "Virus and Cell Specific HMGB1 Secretion and Subepithelial Infiltrate Formation in Adenovirus Keratitis," has been formally accepted for publication in PLOS Pathogens.

Best regards,

Sumita Bhaduri-McIntosh

Editor-in-Chief

PLOS Pathogens

orcid.org/0000-0003-2946-9497

Michael Malim

Editor-in-Chief

PLOS Pathogens

orcid.org/0000-0002-7699-2064